# Human navigation strategies and their errors result from dynamic interactions of spatial uncertainties

Fabian Kessler ●[1] ✉, Julia Frankenstein ●[1] & Constantin A. Rothkopf ●[1,2]

Goal-directed navigation requires continuously integrating uncertain self-motion and landmark cues into an internal sense of location and direction, concurrently planning future paths, and sequentially executing motor actions. Here, we provide a unified account of these processes with a computational model of probabilistic path planning in the framework of optimal feedback control under uncertainty. This model gives rise to diverse human navigational strategies previously believed to be distinct behaviors and predicts quantitatively both the errors and the variability of navigation across numerous experiments. This furthermore explains how sequential egocentric landmark observations form an uncertain allocentric cognitive map, how this internal map is used both in route planning and during execution of movements, and reconciles seemingly contradictory results about cue-integration behavior in navigation. Taken together, the present work provides a parsimonious explanation of how patterns of human goal-directed navigation behavior arise from the continuous and dynamic interactions of spatial uncertainties in perception, cognition, and action.

Navigation is one of the most fundamental goal-directed behaviors relevant to survival in animals and humans. As navigation is the cognitive ability to both plan and move from one place to a goal location[1], it involves dynamic interactions of multi-sensory perceptual, cognitive, and motor processes[2]. Human navigation performance has been shown to exhibit large variability[3,4], which has been associated with noise in sensory[5], representational[6], and motor systems[7]. However, while previous research has investigated navigation behaviorally[3], its neuronal underpinnings[8], prominently internal representations related to a cognitive map[9], and through computational modeling[10,11], a comprehensive account of uncertainty and variability involved in navigation is still elusive[12]. Understanding how these systems interact and how navigational variability is caused by noise in these systems is crucial for understanding how the brain makes use of ambiguous, uncertain, and noisy information from different, sometimes conflicting, sources[13] to maintain an inner sense of location and direction while moving through the environment.

The paradigmatic experiment for studying how humans make use of information in navigation is the triangle-completion (or homing) task[3], in which participants navigate a triangular-shaped outbound path by picking up a sequence of three objects in a real[14] or virtual environment[15,16] with landmarks prior to returning to the location of the first object. This task and its variants allow manipulating the availability of different sources of information, including internal, e.g., motor cues[7] or external sources, e.g., landmarks[17]. Perhaps surprisingly, this task is sufficient to elicit a broad range of behaviors, including path-integration[3], beaconing[18], landmark-based strategies[17,19], strategies based on route knowledge[20], or the use of a cognitive map[21]. However, just the question of whether information from landmarks and path-integration are combined when estimating the position or homing direction in navigation[14–16,22] is still unresolved. Thus, it is still unknown why and under which circumstances these behaviors are adopted[12] and how this might relate to the different sources of uncertainty, which fundamentally requires recurring to a computational model.

[1]Centre for Cognitive Science & Institute of Psychology, Technical University of Darmstadt, Darmstadt, Germany. [2]Frankfurt Institute for Advanced Studies, Goethe University, Frankfurt, Germany. ✉e-mail: fabian.kessler@tu-darmstadt.de

Although numerous computational models of navigation have been developed, the vast majority of these models do not involve uncertainty[10,11]. Models involving sensory uncertainties integrate two static sensory cues, i.e., landmarks and self-motion, but neither account for internal representational nor motor uncertainty and provide puzzling and contradicting conclusions regarding empirically observed behavior[14–16]. A notable recent exception involves dynamic sensory uncertainty[23]. But, more importantly, all these models are ideal observer models, considering a single percept or action, i.e., selecting a single response location[14,16] or direction[15]. However, navigation is inherently a sequential visuomotor behavior, which unfolds over space and time and requires the actor to continuously and dynamically integrate sensory uncertainty, internal model uncertainty, motor variability, and behavioral costs to plan subsequent actions[12,24–26]. Thus, this suggests adopting the framework of optimal feedback control

under uncertainty[27], which has been able to explain a vast array of natural sensorimotor behaviors, including basic motor control tasks[28–30], locomotion[31], and ball-catching behaviors[25].

Here, we hypothesized that the broad range of behavioral phenomena in human goal-directed navigation arises from the sequential and dynamic interaction of uncertainties in perception, internal representations, and actions as predicted by optimal feedback control under uncertainty. To test this hypothesis, we formulated a dynamic Bayesian actor model of navigation in the framework of optimal feedback control under uncertainty assuming that humans continuously and sequentially plan their navigation actions based on subjective, internal beliefs combining different, uncertain sources of information. The model accounts for state estimation (Where am I? Where is my goal?), learning (What is the layout of the environment?), as well as path planning and control (Where should I go? How do I get

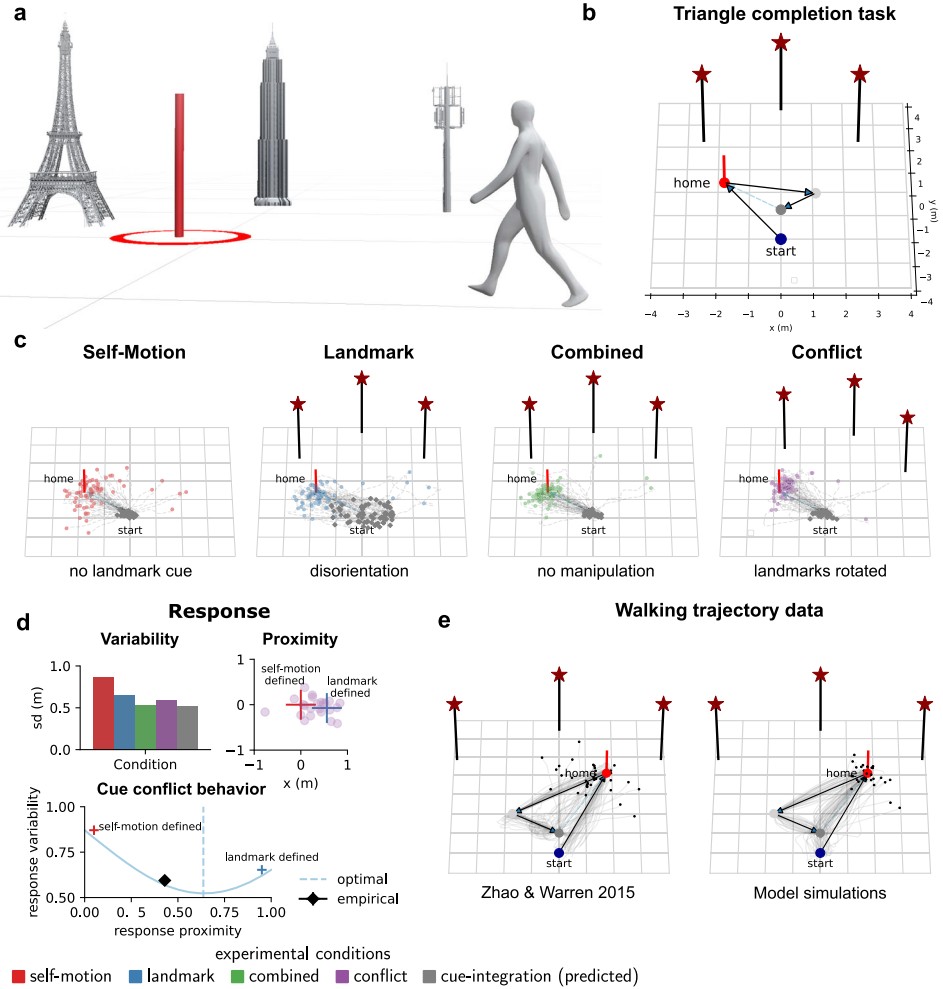

**Fig. 1 | Triangle-completion task & cue integration account. a** Schematic of a typical homing task, here triangle completion, within a real[14] or virtual[15,16] environment with landmarks in the background and a goal location (shown in red). **b** Participants walk from a start position through a sequence of three-goal locations (outbound path) before returning to the first goal location (home, shown in red). **c** Experimental manipulations of internal and external cues at the end of the outbound path for different experimental conditions: Self-motion condition (reduced visual information), landmark condition (reduced direction and position information), combined condition (all cues available), conflict condition (covert rotation of landmarks). Endpoint data from Chen et al.[16]. **d** Cue integration accounts for endpoint variability in homing task[14]. Response variability is computed as the standard deviation of Euclidean distances[14,16] (or heading direction[15]) to mean response locations (or directions) for each condition. Top-Left: Response variability for

different cue conditions. Variability in landmark (blue) and self-motion (red) conditions predict reduced variability in combined conditions (green and violet) according to perceptual cue integration models (gray). Top-Right: Biased homing responses in conflict conditions are used to determine how much participants relied on either of the two cues (relative response proximity). Red and blue crosses represent target locations for exclusive reliance on either self-motion or landmarks cues. Bottom: If cues are combined optimally in conflict conditions, response variability is reduced in double cue conditions (combined and conflict) compared to single cue conditions (landmark and self-motion) and the optimal response location is biased toward the more reliable cue. **e** Walking trajectories of participants from the study of Zhao & Warren[15] compared to simulated participants from our computational model performing the triangle-completion task (Supplementary Movie 2).

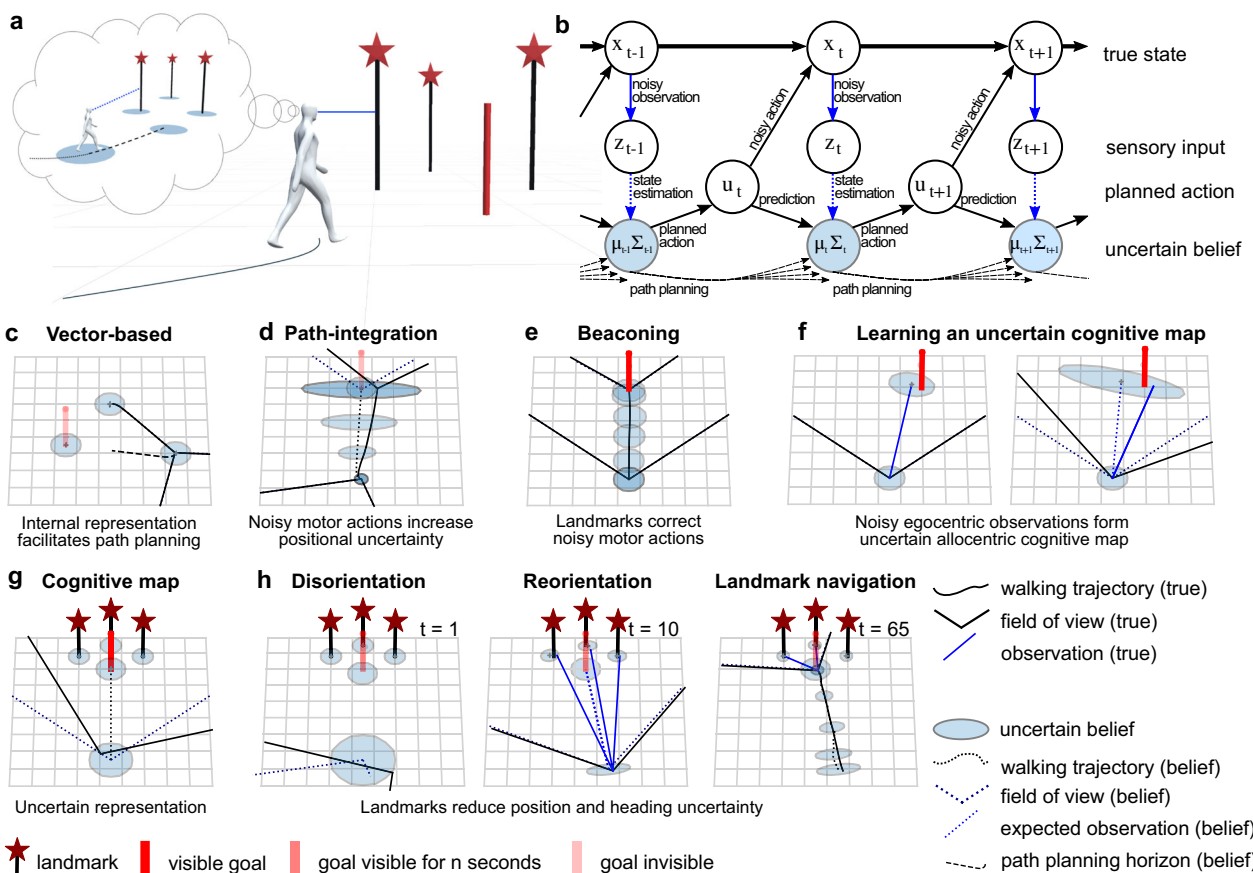

**Fig. 2 | Diverse navigational strategies dynamically emerge in simulations from the interaction of perceptual and internal model uncertainties during optimal control under uncertainty. a** Participants moving in an environment with landmarks and target locations build up a probabilistic allocentric internal representation by sequentially combining egocentric observations, capturing their uncertain subjective belief regarding their position, heading direction, as well as the expected location of landmarks and goals. **b** The true state $x_t$ which captures participant's location, heading direction, as well as the location of goals and landmarks, is not directly accessible, hence participants rely on their probabilistic internal representation $b_t = (\mu_t, \Sigma_t)$, reflecting their subjective spatial uncertainties, which serves as the basis for path planning and is updated by planned motor actions $u_t$ (prediction) alongside noisy sensory observations from perception $z_t$ (state estimation). **c** The internal subjective belief facilitates the planning of trajectories towards internally represented goal locations and sequential updating of the participant's allocentric position within the internal representation (vector-based navigation). A goal is considered to be reached by a participant when the belief about self-location matches the belief of the goal location in the internal representation. **d** In the absence of landmarks, noisy motor actions increase positional and heading uncertainty tracked by the belief state, which may become increasingly discrepant from the actual state $x_t$ (path-integration). **e** During locomotion towards a visible goal, visual feedback from sensory perception reduces positional errors stemming from noisy motor actions (beaconing). **f** Initial noisy egocentric observations of objects (distance and bearing) are transformed into an internal allocentric representation taking into account the participant's positional and observation uncertainties (left: oriented, low uncertainty, right: disoriented, high uncertainty). **g** The allocentric internal representation, reflects participant's spatial estimates and uncertainties regarding the location of landmarks and goals in the environment, as well as their own location and heading direction within the environment (cognitive map). **h** Landmarks in the environment reduce participant's positional and heading uncertainty and provide orientation (landmark-based navigation). Left: Participant is disoriented, i.e., belief about heading and position is discrepant from true heading and position and highly uncertain. Middle: Landmark observation reduces positional uncertainty allowing participants to re-orient. Right: Movement toward the invisible goal location using landmark observations to correct errors due to noisy motor actions.

---

there?) while jointly considering uncertainties in perception, representation, and action. Importantly, the model takes into account the well-known facts that human sensory uncertainty is state-dependent[5], internal motor variability is signal-dependent[32], and uncertainty of the internal map is time-dependent[33]. Recent advances in probabilistic planning[34] efficiently allow computing path planning and navigation behavior.

The results provide a unifying account of goal-directed navigation as different behaviors[1], previously assumed to be distinct strategies[19], arise from differential interactions of uncertainties under different environmental conditions. The errors and the variability of different navigation behaviors[14] are well predicted across multiple experiments and can be related to the dynamically interacting uncertainties and variabilities that accrue over space and time. Finally, this explains a broad range of seemingly contradictory phenomena, such as assumed violations of cue integration reported in previous work[15,16]. Our results emphasize the fundamental role of the dynamic evolution of uncertainties during goal-directed navigation, which the actor shapes actively through behavior, demonstrating that perception, cognition, and action in spatial navigation must be considered jointly because they are inseparably intertwined[26].

## Results
### Triangle-completion task with landmarks
To better understand how humans dynamically utilize internal and external cues during goal-directed navigation, we obtained the environmental layouts, methodological details, and behavioral data from three previously published studies carried out in different

laboratories[14–16] (see "Methods" section for more detail; Supplementary Fig. 1 and Supplementary Table 1). In all studies, human participants performed a triangle-completion task in either a real or a virtual environment containing landmarks (Fig. 1a). On each trial, participants viewed the environment from their starting position. They then walked an outbound path through a sequence of three-goal locations (Fig. 1b). Once they reached the last target location, they had to wait 8-20 seconds, during which the environment was rendered invisible. When the environment reappeared, internal or external cues were manipulated under one of four experimental conditions and participants had to return to the remembered first goal location leading to different patterns of endpoint variability (Fig. 1c). For comparison between empirical data and data simulated from our computational model, we replicated analyses procedures from previous studies including predictions of the cue integration model[14–16] (Fig. 1d and see "Methods"). Full statistics for all main analyses are provided in the Source Data file.

## Navigational strategies emerge from optimal control's dynamic interactions of spatial uncertainties

The homing task requires participants to walk towards visible and invisible goals while keeping track of their position, learning an internal representation of the environment consisting of landmarks and goal locations, (re)orienting based on known landmarks, and finally returning to the remembered home location. Crucially, motor, perceptual, and representational processes are inherently noisy (see Supplementary Fig. 2), which implies that the participants's actual position and heading direction are unknown to them during navigation. Both can only be inferred from noisy egocentric observations in combination with a learned uncertain allocentric representation serving as a cognitive map[23]. We therefore suggest that the brain utilizes this internal representation for sequential learning, path planning, and inference while accounting for the inherent uncertainties in perception, representation, and action across space and time. Here, we provide an account of how this can be achieved computationally.

To better understand how humans make use of ambiguous, uncertain, and noisy information from different, sometimes conflicting, sources of spatial information, we frame goal-directed navigation in terms of optimal control under uncertainty[27]. This allows us to describe how spatial uncertainty in the internal representation arises from the perspective of participants moving through the environment and how this uncertainty shapes goal-directed navigation behavior and navigational strategies under different experimental and environmental manipulations. A detailed mathematical description of the model's properties, e.g., motor actions, landmark perception, internal representation, path planning, and the choice of model parameters, is provided in "Methods" and the accompanying model[35].

In the homing task, the state of the world $x_t$ at a particular time $t$ describes the participants's true physical position, heading direction, and the position of landmarks and goals in the environment. Participants moving through the environment execute noisy motor actions $u_t$ consisting of changes in linear and angular velocity subject to signal-dependent noise and obtain noisy egocentric sensory input $z_t$ of the distance and direction of landmarks subject to state-dependent noise. Importantly, we assume that participants have access to an integrated percept stemming from different visual depth cues and an integrated multi-sensory signal of self-motion cues.

To effectively deal with the growing spatial uncertainty resulting from these noisy inputs, participants maintain a dynamic belief distribution $b_t = (\mu_t, \Sigma_t)$, i.e., a belief state, over possible world states[24]. Specifically, this belief represents a probability distribution for the participant to be at a specific location and orientation in space along with the locations of landmarks and goals in the environment, thus effectively serving as an uncertain internal map[12,24] (Fig. 2a). The participants' internal belief is continuously updated by an integrated sensory input from external cues and motor actions via an internal

model of the underlying dynamics, which allows for inferring allocentric position from noisy egocentric sensory input and self-motion signals (Fig. 2b). The magnitude of noise in these cues was manually constrained offline using findings from previous experiments, resulting in physically, biologically, and cognitively plausible ranges (see Supplementary Table 2 and Supplementary Fig. 3a–i). We also assumed these parameters to be equal for both the task's generative model and the participants' internal uncertainties.

During path planning, motor actions $u_t$, i.e., forward locomotion and change of heading direction, are selected by trading off costs for goal reaching, control effort, and the accrual of uncertainty over the current position, heading direction as well as over the location of landmarks and goals. Importantly, because sequential actions are selected based on uncertain internal beliefs, planning accounts for both actions that bring humans closer to their navigational goal and actions that reduce uncertainty about the current position, direction, and environment layout. This is the reason, why, perhaps, surprisingly, diverse navigational strategies dynamically emerge from this optimal control under uncertainty formulation, depending on the availability of spatial cues and the relative uncertainties within the internal representation (Supplementary Movie 1). Thus, this brings navigational strategies into a unifying framework and shows that path planning based on internal subjective beliefs explains how humans actively shape perceptual and internal uncertainties through navigational behavior.

Specifically, the internal representation facilitates the planning of direct paths toward goals, a strategy known as vector-based navigation[11] (Fig. 2c). However, in the absence of landmarks, noisy motor actions update the participants' positional and heading belief, while increasing uncertainty, which is captured by the belief state tracking the current location and heading direction (Fig. 2d), a strategy referred to as path-integration[4,19,36]. Accordingly, due to signal-dependent noise in motor systems, the participants' belief about the position and heading may become increasingly more discrepant from the participants' actual position and heading direction, such that participants eventually become disoriented, leading to highly variable navigation errors commonly associated with sole reliance on path-integration[3,7,36].

To remain oriented, humans rely on landmarks in their immediate environment[18,19], as they provide information in terms of their relative distance and direction and can serve as visual beacons during goal-directed movement, i.e. beaconing (Fig. 2e). For landmarks to provide useful information about one's location and orientation within the environment, their location needs to be internally represented first[18,37], thus requiring sequential learning of an internal allocentric representation from egocentric observation[38]. However, due to dynamic interactions of perceptual, motor, and representational variabilities this internal representation, akin to a cognitive map, is fundamentally uncertain (Fig. 2f). For example, the degree of uncertainty over a landmark's location depends on the participant's current positional and heading uncertainty at the time of observation as well as the perceptual uncertainty, which differs depending on the distance and eccentricity of the landmark relative to the observer. Landmarks and other locations already stored in the internal representation are subject to representational variability; thus, uncertainty over their position increases slowly over time (Fig. 2g).

Once the location of a landmark or any other object is stored in the internal representation, it provides participants with orientation within the environment. A participant is disoriented if their true location and heading direction are discrepant from the internally represented heading direction and location, and they maintain a high degree of spatial uncertainty regarding these quantities. If the goal location is not visible but landmarks are present, they re-orient a disoriented participant by providing an estimate of the current position and heading direction relative to the internal allocentric

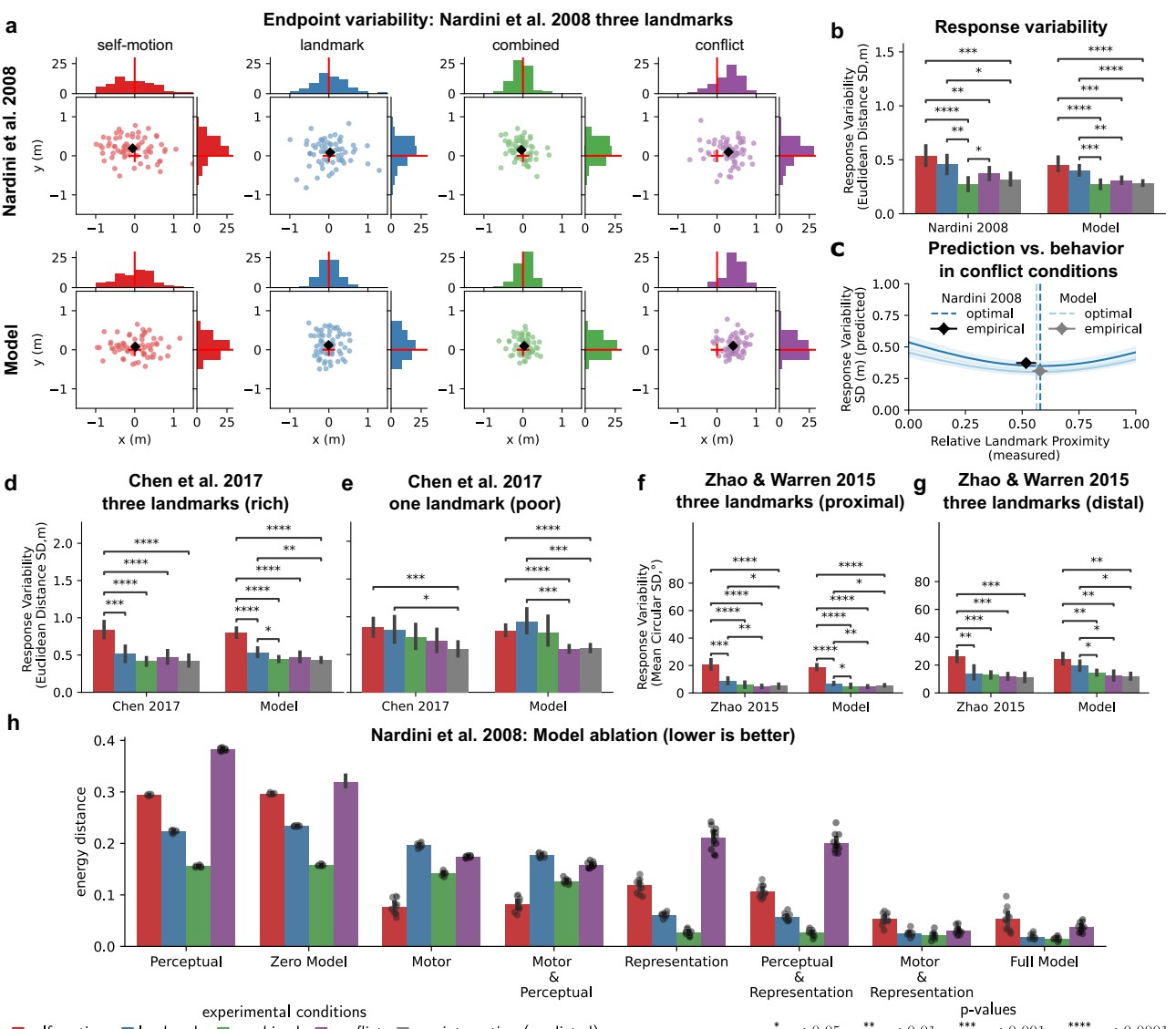

**Fig. 3 | Noise in perception, representation, and action explains endpoint variability across multiple studies and experimental manipulations. a** Endpoint distributions for all four conditions for (simulated) participants from Nardini et al.[14]. Homogeneity energy test: self-motion ($p = 0.296$), landmark ($p = 1.0$), combined ($p = 0.196$), conflict ($p = 0.156$). **b** Analysis of response variability and cue integration for (simulated) participants from Nardini et al.[14] ($n = 17$). A two-way repeated measures ANOVA testing type (model vs. empirical) and cue condition (self-motion, landmark, combined, conflict) showed a statistically significant effect for condition ($F_{(4, 128)} = 25.83$, $p < 0.0001$, $\eta^2 = 0.31$), but not for type ($F_{(1, 32)} = 2.88$, $p = 0.1$, $\eta^2 = 0.03$), with no significant interaction ($F_{(4, 128)} = 0.7$, $p = 0.6$, $\eta^2 = 0.01$). Post-hoc $t$-test (two-sided). Error bars represent mean response variability $\pm 1$ SD. **c** Model predictions vs. actual behavior in conflict conditions for (simulated) participants from Nardini et al.[14]. Error bars indicate relative landmark proximity ($x$-direction) and response variability ($y$-direction) with $\pm 1$ SEM. **d, e** Analysis of response variability across cue conditions for (simulated) participants from Chen et al.[16] ($n = 18$). In the three-landmark environment, statistically significant main effects were found for cue condition ($F_{(4, 136)} = 75.22$, $p < 0.0001$, $\eta^2 = 0.48$), but not for type ($F_{(1, 34)} = 0.01$, $p = 0.94$, $\eta^2 = 0.0$), with no significant interaction

($F_{(4, 136)} = 0.59$, $p = 0.67$, $\eta^2 = 0.004$). In the single landmark environment, the main effects were statistically significant for cue condition ($F_{(4, 136)} = 12.36$, $p < 0.0001$, $\eta^2 = 0.139$) without significant effects for type ($F_{(1, 34)} = 0.01$, $p = 0.94$, $\eta^2 = 0.0$) or their interaction ($F_{(4, 136)} = 1.28$, $p = 0.28$, $\eta^2 = 0.014$). Post-hoc $t$-test (two-sided). Error bars represent mean response variability $\pm 1$ SD. **f, g** Response variability across cue conditions for (simulated) participants from Zhao & Warren[15]. In the proximal landmark environment ($n = 6$), statistically significant main effects for cue condition were observed ($F_{(4, 40)} = 81.91$, $p < 0.0001$, $\eta^2 = 0.86$), with no significant effects for type ($F_{(1, 10)} = 3.23$, $p = 0.1$, $\eta^2 = 0.006$) or their interaction ($F_{(4, 40)} = 0.59$, $p = 0.67$, $\eta^2 = 0.01$). In the distal landmark environment ($n = 5$), there was a significant main effect for cue condition ($F_{(4, 32)} = 31.37$, $p < 0.0001$, $\eta^2 = 0.636$), but not for type ($F_{(1, 8)} = 0.59$, $p = 0.46$, $\eta^2 = 0.011$) or their interaction ($F_{(4, 32)} = 2.05$, $p = 0.11$, $\eta^2 = 0.042$). Post-hoc $t$-test (two-sided). Error bars represent mean response variability $\pm 1$ SD. **h** Impact of perceptual, representational, and motor variability on endpoint distributions in the homing task from Nardini et al.[14]. Energy distances were calculated between empirical and simulated data ($n = 10$) for each condition (Supplementary Fig. 4). Error bars indicate mean energy distance $\pm 1$ SD. Source data are provided as a Source Data file.

representation[37] while also reducing the participant's spatial uncertainty, which then facilitates reliable path planning. Subsequently, the landmarks aid in reducing motor errors, and the resulting positional and heading uncertainty caused by noisy motor actions, when navigating toward the remembered goal location (Fig. 2h), a strategy referred to as landmark-based navigation[17,19,39].

When faced with multiple cues and, therefore, multiple potential strategies, a considerable challenge lies in effectively arbitrating between these strategies. Previously, it has been suggested that navigators either switch between various navigation behaviors[19] or integrate them[40,41]. In this process, differences in uncertainties related to representational, motor, and perceptual processes mediate the

integration of different strategies, shaping the resultant navigational behavior and its variability. Intuitively, these uncertainties correspond to the fidelity of the navigating agent's allocentric knowledge of the surrounding environment (representation), their ability to update and maintain a sense of self-location in response to noisy actions (motor), and their ability to egocentrically perceive the distance and direction of objects in the environment required for both map-learning and self-localization (perceptual). In the presence of multiple cues and potential strategies, differing levels of navigational variability may emerge more subtly and continuously. Specifically, relative differences in internal uncertainties related to sensory, motor, and representational processes continuously shape the balance of how much cognitive map-learning, self-localization, and internal prediction from self-motion take place during navigation, which consequently affects navigational precision.

## Endpoint variability in goal-directed navigation arises from noise in perception, internal representations, and action

The dynamic Bayesian actor model allows simulating behavior for different experimental conditions in previous studies[14–16] and obtaining walking trajectories for homing tasks (Fig. 1e and Supplementary Movie 2). These studies have shown that repeatedly walking back to a goal location leads to variability in the endpoints of walking trajectories, which depends on the availability and reliability of cues prior to and during the homing response. However, currently, it is unknown how or why this behavioral variability arises and how self-motion and landmark cues interact during this process. Here, we show in simulations of homing tasks across five experiments from three different studies and laboratories that the observed endpoint biases and variability are quantitatively predicted by the interaction of three sources of sensory-motor noise, namely perceptual, representational, and motor noise when assuming that humans plan their actions based on subjective internal beliefs.

Simulations of participants by our computational model performing the triangle-completion task within the environment from Nardini et al.[14] (272 trials from 17 simulated participants) under the four homing conditions (self-motion, landmark, combined, conflict) revealed endpoint distributions for which a homogeneity energy test indicated there was no statistically significant difference between conditions (homogeneity energy tests between conditions; all $p > 0.05$; see Fig. 3a). Moreover, the observed standard deviation of Euclidean distances in homing responses, i.e., response variability, revealed no statistically significant differences for data by Nardini et al.[14], simulated participant data and predictions of the cue integration model (Fig. 3b), which predicted that response variability in the combined cue condition is lower than in single cue conditions and relative response proximity to landmark cues in conflict conditions (Fig. 3c).

Keeping all parameters pertaining to the actor model constant, we subsequently simulated behavior for participants performing the homing tasks in four additional experiments from two studies[15,16] with different environmental layouts, differing in the locations of goals as well as the number and locations of landmarks, and experimental manipulations (Supplementary Table 1 and Supplementary Fig. 1). In these studies, goal locations were not visible from the beginning of the task and, therefore, not observable all at once but appearing sequentially after successful completion of each leg of the triangular path.

Simulations of navigation behavior from two experiments by Chen et al.[16], in which humans performed the homing task in either an environment with three landmarks (720 trials from 18 simulated participants) or only a single landmark (720 trials from 18 simulated participants), revealed no statistically significant differences in the distribution of endpoints between empirical data and simulated participants for both environments and across the four cue conditions (homogeneity energy tests between conditions; all $p > 0.05$; see

Supplementary Fig. 5a, b for exact $p$-values). Further, there was no statistically significant difference between empirically observed response variability across cue conditions, behavioral data for simulated participants, and predictions of the cue integration model (Fig. 3d, e). This fits previous observations that the availability of multiple landmarks during homing reduces endpoint variability as they provide relatively more location information than a single landmark[17,39,42].

It is well-known that navigational performance is not only affected by the number of landmarks but also the proximity of landmarks relative to the actor[5,18]. To explicitly test for this prediction, we simulated data from two experiments by Zhao & Warren[15] (proximal: 2160 trials from six simulated participants; distal: 1800 trials from five simulated participants), in which landmarks were either 5.5m or 500m away but in the same relative configuration. Visual inspection yields similar endpoint distributions between empirical data and model simulations for the proximal and distal landmark environments. In particular, conditions with distal landmarks exhibit larger heading and distance errors as they provide less information regarding their distance and direction to the observer compared to conditions with proximal landmarks (Supplementary Fig. 5c, d). Although qualitatively in agreement, quantitative comparison using homogeneity energy tests for differences between empirical and model endpoint distributions reveals statistically significant differences for both the proximal and distal environments (homogeneity energy tests between conditions; all $p < 0.05$; see Supplementary Fig. 5c, d for exact $p$-values). Significant differences between these endpoint distributions could be explained by a larger sample size (3 and up to 7 times as many trials compared to the other experiments). Conversely, analysis of response variability on the participant level reveals no statistically significant difference in heading error magnitudes between empirical data and model simulations for both the environment with proximal landmarks and the environment with distal landmarks (Fig. 3f, g).

To better understand how the dynamic interaction of three different sources of sensory-motor noise is reflected in the observed endpoint variability, we further tested the contribution of perceptual, representational, and motor variability separately by formulating eight different models (see "Methods") in which we selectively excluded specific sources of variability in the generative process and the corresponding internal uncertainties in the actor model. This allowed us to quantify their impact on endpoint variability within the four cue conditions. We compared endpoint distributions simulated from these differing models (10 datasets for each model) against the behavioral data (see Fig. 3h; Supplementary Fig. 4 for exemplary simulated endpoint distributions for behavioral data from Nardini et al.[14]).

These results show that neither perceptual, representational, nor motor variability alone can account for the observed endpoint variability of participants across all four experimental conditions. Overall, participants' decaying memory for landmark and goal locations had the strongest influence on overall endpoint variability across cue conditions, whereas, for example, the influence of perceptual variability was negligible when directly comparing models with and without perceptual variability. Most likely because participants observed the entire environmental layout, including the locations of all three goals and the three landmarks in the background at the beginning of each trial, which allowed them to initially form a comparatively accurate representation of the environment, which then slowly decayed over time.

In the four experiments by Chen et al.[16] and Zhao & Warren[15], however, goal locations appeared sequentially, requiring participants to learn the environmental layout from noisy egocentric observations during the unfolding of the triangle-completion task. Consequently, perceptual and motor noise in these experiments had a comparatively higher contribution to endpoint variability across the different homing conditions as they directly affected the learning of the internal

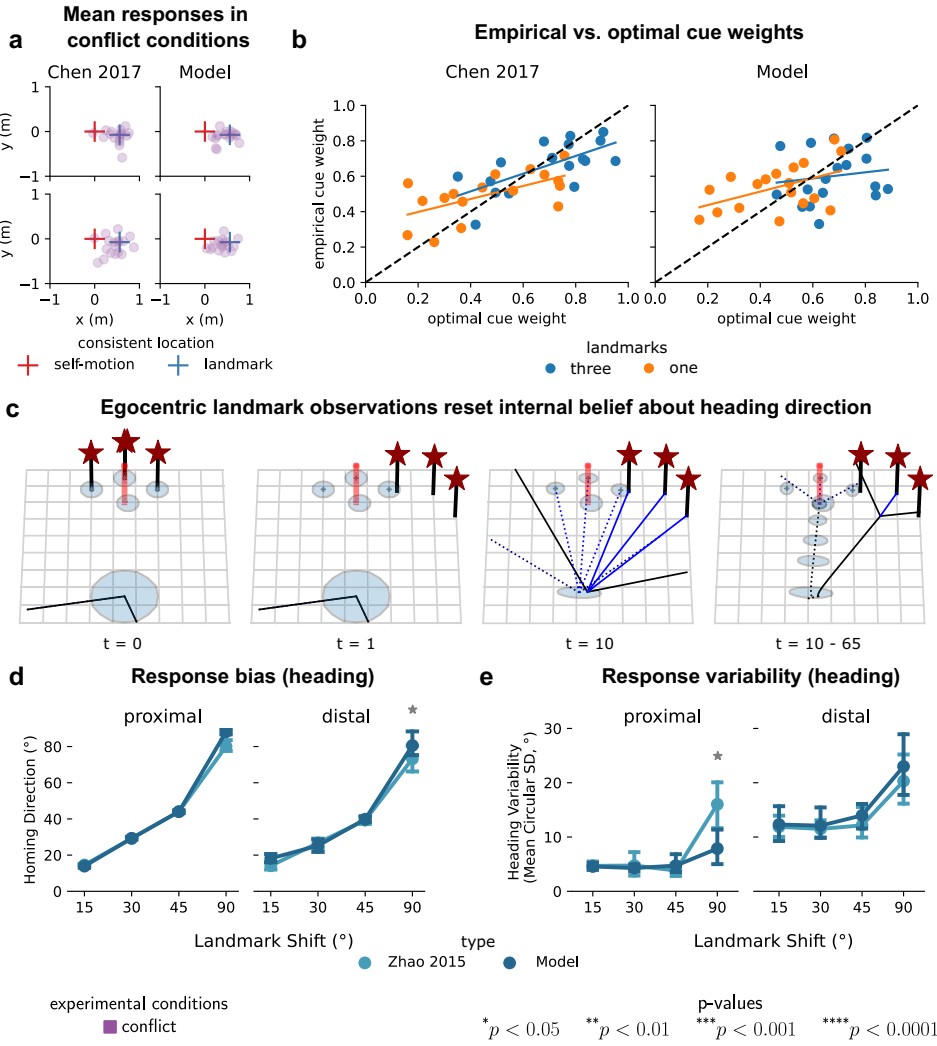

**Fig. 4 | Optimal sequential actions under perceptual, representational, and motor uncertainties predict seemingly sub-optimal cue-integration behavior in conflict conditions. a** Mean responses for conflict conditions for participants from Chen et al.[16] (left) vs. simulated participants from our computational model (right) for an environment with three landmarks (top) and one landmark (bottom). **b** Optimal vs. empirical cue weights for participants from Chen et al.[16] (left) vs. simulated participants from our computational model (right). Dots indicate individual (simulated) participants. Optimal cue weights were computed based on observed response variability in single cue conditions, whereas empirical cue weights were computed based on response proximity to either cue location in conflict conditions. Linear regression lines were fitted to (simulated) participants' cue weights for the two environments separately. **c** The reliance on landmark cues during homing is explained by the integration of egocentric landmark observations with subjective internal beliefs (see Supplementary Movie 4). Initially ($t = 0$), participants hold beliefs about their own location and that of landmarks. At $t = 1$, landmarks are covertly rotated, but participants' beliefs about their expected positions remain unchanged. By $t = 10$, participants turn around, encountering increased uncertainty in heading due to landmark rotation. The sight of landmarks recalibrates their internal heading estimate since they expect landmarks straight ahead, reducing uncertainty in position and heading. From $t = 10$ to $t = 65$, during homing, participants utilize their internal beliefs for path planning, employing landmarks to correct motor errors. This reduces positional uncertainty and variability in endpoints, though homing responses are biased. **d** Response bias in conflict conditions for (simulated) participants from Zhao & Warren[15]. Two-sided paired $t$-tests for angles 15°, 30°, 45°, and 90° with $p$-values for the proximal environment ($n = 5$) of 0.33, 0.93, 0.96, and 0.001, and for the distal environment ($n = 6$) of 0.06, 0.57, 0.71, and 0.25. Error bars show mean homing direction $\pm 1$ SD. **e** Response variability in conflict conditions for (simulated) participants from Zhao & Warren[15]. Two-sided paired $t$-tests for angles 15°, 30°, 45°, and 90° with $p$-values of 0.85, 0.72, 0.46, 0.02 for the proximal environment ($n = 5$) and 0.85, 0.77, 0.41, 0.53 for the distal environment ($n = 6$). Error bars show mean homing variability $\pm 1$ SD. Endpoint distributions are shown in Supplementary Fig. 14. Source data are provided as a Source Data file.

representation and subsequent localization based on this representation (Supplementary Figs. 6–9). Importantly, the relative contributions of the different sources of uncertainty and the observed response variability vary between different environmental layouts and experimental conditions, suggesting that these different sources of uncertainties dynamically interact during the unfolding of the triangle-completion task as opposed to being merely additive.

Models lacking motor variability but incorporating perceptual or representational uncertainty fail to explain observed shifts in homing errors in conflict conditions. This is because the absence of motor variability eliminates movement-related errors, leading to increased positional and heading uncertainty and, consequently, increased reliance on internal predictions from self-motion rather than landmarks for reorienting during homing. Further models lacking perceptual variability perform relatively worse in the distal landmark condition of Zhao & Warren[15] as landmarks 500m away provide less information for reaching a close target in the proximity of 3 to 4 meters. Across all conditions and experiments, representation variability, i.e., the uncertainty in the internal map, remained the most significant influence, and models including both

representation and motor variability provided the closest matching endpoint distributions.

To better understand the importance of particular noise sources in sensory, motor, and representational systems, we conducted additional simulations based on randomly sampled noise parameters, where each parameter was uniformly drawn between 0 and up to one magnitude higher than our selected parameter values (see Supplementary Fig. 3a–i). Model simulations using 1000 randomly sampled noise parameter sets reveal that the model is highly sensitive to the choice of particular noise parameters (see Supplementary Fig. 11a) and that model simulation generating endpoint distributions more closely resembling empirically observed data are significantly influenced by particular noise parameters (see Supplementary Fig. 11b). Conversely, the model showed less sensitivity to variations in planning and control horizons (see Supplementary Fig. 10a, c). While adapting planning horizons is crucial in unpredictable environments and scenarios like uneven terrains[43] or obstacle avoidance[44], the homing task did not involve such unpredictability. Similarly, for the homing task by Nardini et al.[14] the model was less sensitive to precise choice of cost-function weights (see Supplementary Fig. 10e), complementing previous findings which show that difference in cost-function weights primarily influence the shape[45] and velocity profiles[46] of trajectories, but not necessarily their endpoints.

Considering the potential significance of specific sources of uncertainty over others, we asked whether models with fewer sources of variability would provide similar or even superior predictions compared to a model that accounts for all sources of variability. However, due to the considerable computational demands of model simulations (requiring 5-10 minutes per trial), we could not perform an exhaustive model comparison where each model is separately optimized to provide the closest matching endpoint distributions. Therefore, we performed a random model comparison, where each of the eight models was simulated under 50 randomly sampled noise parameter sets. A closer analysis of energy distances for each of the eight models across the different experimental conditions suggests that some models are inherently limited in their expressiveness (Supplementary Fig. 11c). For example, the zero, perceptual, and motor models show little to no variation in energy distances across the different conditions, irrespective of specific parameter values. Similarly, models that incorporate uncertainty in perception, representation, or both but lack motor variability are limited in their expressiveness in the conflict condition.

Next, we chose the best-performing random parameter set for each model. We then subjected them to transfer testing, where we simulated each model for all five experiments to assess their ability to generalize across these varied experimental environments and manipulations. For empirically observed endpoint variability, we found that while some models performed similarly or better than the model with our selected parameter set in certain experiments, overall, their generalization was inferior across all experiments and environments (see Supplementary Fig. 13). Importantly, the parameters in the alternative models found using this procedure would require assigning noise parameters that are up to one order of magnitude higher compared to models excluding particular sources of uncertainty to compensate for the absence of variability in other sources. This leads to parameter magnitudes that are physically, biologically, and cognitively rather implausible (see Supplementary Fig. 3j). Additionally, for most models, this leads to walking trajectories that do not align well with empirically observed trajectories (see Supplementary Fig. 12).

Taken together, these results show that optimal control under perceptual, representational, and motor uncertainties quantitatively predicts and qualitatively explains human navigation behavior across multiple studies with different environmental configurations and manipulations and matches predictions of cue integration models[14–16]. Specifically, the dynamic evolution of spatial uncertainties explains how endpoint variability and bias arise within the four different homing conditions from sequential interactions of perceptual, representational, and motor variabilities (Supplementary Movie 3).

## Cue weighting between self-motion and landmarks arises dynamically from sequential interactions of perceptual, representational, and motor uncertainties

Previous studies on cue combination in navigation investigated to what degree humans rely on self-motion information relative to the information provided by landmarks during homing and posited that humans adjust their weighting optimally based on relative cue reliability[14–16]. However, the reliability of self-motion and landmark cues was derived from observed variability of the walking trajectory endpoints in the respective single cue conditions and then used to compute optimal cue weights according to their relative reliability (see "Methods"). In conflict conditions, during which both types of cues are present but put in conflict unbeknownst to participants by rotating landmarks 15° around the location of the last target[14], optimal cue weighting should be reflected in participants' reliance on landmark cues compared to self-motion cues. Particularly, whether participants' responses followed the self-motion-consistent or the landmark-consistent location or fell somewhere between the two locations. This response proximity to either cue (Fig. 4a) allows for computing empirical cue weights, with zero indicating exclusive reliance on self-motion cues, and one exclusive reliance on landmark cues.

The study by Chen et al.[16] reported that the cue integration model could not explain the discrepancy between optimal cue weights (computed from the variances of walking endpoint distributions in single cue conditions) and empirical cue weights (computed from the means of response distributions in conflict conditions) and interpreted this as sub-optimal behavior. For optimal behavior, Chen et al.[16] proposed a perfectly linear relationship between both types of cue weights across participants and both environments (Fig. 4b, left). However, analysis of regression coefficients shows intercepts larger than zero and slopes less than one, indicating a regression to the mean rather than the proposed perfectly linear relationship (one landmark: $R^2 = 0.361$, $p = 0.011$, slope 95%-ci (0.099, 0.636); three landmarks: $R^2 = 0.45$, $p = 0.003$, slope 95%-ci (0.195, 0.804)). When interpreting these findings through the lens of the optimal cue integration model, participants overweighted the influence of the self-motion cue and underweighted the influence of the landmark cue, compared to what would be predicted from empirical response variability in single cue conditions alone[16,47].

Here we show that endpoints of walking trajectories dynamically arise from the interaction of perception, internal representations, and action instead of as static perceptual cue weights and therefore must not necessarily conform to predictions by the cue integration model[14], which initially was derived for the case of two sensory cues and their empirically measured uncertainties as derived from patterns of endpoint variability in different experimental conditions. Accordingly, we computed both types of cue weights for 18 simulated participants from simulated endpoint data (Fig. 3d and Supplementary Fig. 5). Visual inspection of optimal vs. empirical cue weights yields that these are similar in magnitude between participants from Chen et al.[16] (Fig. 4b, left) and the simulated participants (Fig. 4b, right). One exception is that the empirical data from real participants exhibited more variability between optimal cue weights, compared to our models' simulated participants, as they were all simulated from the same model parameters. Linear regression analysis showed slopes smaller than one and intercepts larger than zero (one landmark: $R^2 = 0.230$, $p = 0.044$, slope 95%-CI (0.011, 0.751); three landmarks: $R^2 = 0.021$, $p = 0.576$, slope 95%-ci (−0.452, 0.796)), indicating a similar regression to the mean effect in model simulations, as reported in Chen et al.[16].

Taken together, these results show that uncertainties about landmarks and path-integration are dynamically integrated during navigation in the continuous Bayesian state-estimation process in which internally represented spatial uncertainties about self-location, heading direction, and the location of landmarks interact, giving rise to the observed behavioral variability and errors. Thus, the distributions of navigational endpoints are not cue reliabilities per se but instead arise from the continuous and sequential interaction of uncertainties in perception, cognition, and action. Specifically, spatial uncertainties dynamically arise during navigation and are involved in planning appropriate motor actions, updating internal beliefs by predicting the consequences of noisy motor actions and making uncertain egocentric observations of landmarks and goals. Initially, this contributes to learning landmarks' and goals' locations and subsequently allows for inferring the allocentric position and heading direction. This provides a comprehensive and succinct computational account of the interaction of uncertainties in navigation, showing that the reliance on self-motion and landmarks is a continuous dynamic process dependent on the involved uncertainties in sensory, motor, and representational systems.

### Landmark observations reset the internal estimate of the heading direction if cue conflicts suggest a model mismatch

For landmark conflicts larger than 15°, Zhao & Warren[15] observed an incongruity between heading variability and heading direction in trajectory endpoints that could not be explained by the previous cue integration model of navigation. Static perceptual cue integration proposes that cues are weighted according to their relative reliability leading to reduced response variability and a response direction that is biased toward the more reliable cue[14]. However, in the study by Zhao & Warren[15] the observed response direction during homing in conflict conditions was almost entirely dominated by a single cue, i.e., the shifted landmarks, while response variability was reduced simultaneously, violating the predictions from static cue integration of two sensory cues[48]. If participants had integrated these cues optimally their response direction would have been a weighted average according to the relative reliabilities observed in single cue conditions[14].

Simulations from our computational model offer an alternative view of this seemingly sub-optimal cue integration behavior by jointly and sequentially considering the effects of uncertainty in motor actions, perception, and internal representations during homing. Specifically, this phenomenon arises due to a mismatch between participants' expectations based on their internal map, their current uncertainty regarding heading direction, and noisy egocentric observation (Fig. 4c and Supplementary Movie 4). Because changes in heading directions are subject to signal-dependent noise, participants turning around to face the landmark during homing will inevitably increase their heading uncertainty. To reduce the resulting heading uncertainty and determine their current heading direction, participants combine noisy egocentric sensory observations of the landmarks' distance and direction with prior expectations of the landmarks' distance and direction based on their internal map. Crucially, because the landmarks have been rotated unbeknownst to them, and thus initially the landmarks' internally represented positions and uncertainties remain unchanged, this causes participants to consider their subjective heading direction to be straight ahead, therefore creating a conflict between participants' subjective and true heading direction. Thus, landmarks re-orient participants by resetting their internal estimate of heading direction and reducing heading uncertainty. During subsequent homing, the landmarks then aid participants in reducing positional and heading uncertainty, which would accumulate in their absence, leading to reduced response variability, although homing responses are biased.

Simulations for conflict conditions of varying degrees between 15° and 90° exhibited biased homing direction (Fig. 4d and reduced response variability (Fig. 4e and Supplementary Fig. 14). There was no statistically significant difference between the model and empirically observed behavior[15] for conflict angles of 15°, 30°, and 45°. At a conflict angle of 90°, a discrepancy between our model and empirical data can be observed, as participants clearly seemed to notice the conflict on some trials, causing them to ignore landmarks cues altogether. Instead, participants navigated toward the internally represented target location with noisy motor actions, leading to high positional and heading uncertainty and consequently higher response variability, i.e., path-integration (Fig. 2b). As this behavior was observed for the conflicts larger than 90° for most participants and trials, we excluded these angles from the present analysis. Such discrete switches in behavior can, in principle, be explained by additional assumptions about trial-by-trial learning and decision rules for the large discrepancy between the expected position of landmarks given participants' uncertain internal belief about heading direction and their noisy observed heading direction relative to landmarks, e.g., causal inference[49].

Taken together, these results show that dominance of landmark cues and simultaneous reduction of response variability in conflict conditions, a behavior previously deemed sub-optimal under cue integration models when considering only empirically observed end-point variability[48], can be explained by an internal model mismatch, in which participants expect the landmarks to be straight ahead based on their internal representation, and consequently reset their internal belief regarding their heading direction. Note, however, that this behavior simply arises in the optimal control under uncertainty model from the Bayesian integration of the involved uncertainties. Crucially, this highlights the importance of considering how internal spatial uncertainties dynamically interact with perceptual and motor uncertainties and how this interaction shapes observable navigation behavior.

## Discussion

Navigating from one place to another requires utilizing information from spatial cues, both internal to the actor and external from the environment, representing them internally, and planning and executing motor actions sequentially. While prior research has shown that humans can combine different sources of spatial information to reduce their navigational variability[14,16,47,50], it is still unclear how or why the observed variability in trajectories and endpoints arises from the perspective of an actor moving through the environment and how self-motion and landmark cues interact in this process[12].

Here, we hypothesized that goal-directed navigation can be understood as optimal feedback control under uncertainty, which implicitly solves the computational problems of spatial localization[5,37,39], learning of an internal representation[21,38], path planning[51], and locomotor control[31]. Thus, this provides a succinct and complete probabilistic model of goal-directed navigation in open-field environments in a single computational framework. We suggest that maintaining an estimate of spatial uncertainty about one's location and heading direction relative to an uncertain internal map when navigating through the environment is crucial, as these quantities are unknown and can only be inferred from noisy sensory data[23].

Specifically, we assume that during goal-directed navigation, humans continuously plan their actions based on subjective, uncertain internal beliefs about where they are and where the objects and landmarks in the environment are. These uncertain internal beliefs initially arise from dynamic sequential interactions of noisy motor, perceptual, and representational processes. Simulations of goal-directed navigation in homing tasks with a single set of parameters for multiple experiments from three different laboratories show end-point distributions in different cue conditions closely resembling those of human participants. This provides a unifying account of the emergence of navigational strategies, errors, and variability in

trajectory endpoints and explains several seemingly contradictory phenomena reported in previous work[15,16].

Importantly, biases in trajectories and their endpoints observed in previous work are better accounted for as arising sequentially from the dynamic interactions of internal spatial uncertainties with perceptual and motor processes during online navigation rather than ideal observer models with static cues and ad-hoc assumptions about spatial priors or explicit cost-functions[47]. This demonstrates that humans perform close-to-optimal navigation under uncertainty across different environmental conditions. While previous ideal observer models of navigation[14–16,23,47], neither include uncertain internal representations, path planning nor motor actions, the present model allows attributing endpoint variability to these respective sources of uncertainties. Selectively excluding individual sources of uncertainty in model simulations provided consistently larger deviations in endpoint distributions, demonstrating the necessity to include all sources of uncertainties and their interaction[4].

Navigation strategies have broadly been categorized based on sensory cues, representation types, and computational mechanisms, mainly differentiating between response-based and place-based strategies[2,41,52]. These strategies are commonly associated with different neural systems, such as the hippocampal and striatal systems, and their selection is influenced by uncertainty-related factors[52,53]. Here, we specifically focused on place-based navigational strategies within open-field tasks[19] involving continuous motor actions and sensory observations, in contrast to response-based maze tasks requiring discrete decision-making actions, which may become habitual through repetition[53].

In this context, the homing task with landmarks elicits several navigation behaviors previously described as different and seemingly separate navigational strategies[3,19–21]. We demonstrate that these strategies, whether they involve walking towards a visible target location (beaconing[18]), continuously updating self-location based on internally generated motor cues (path-integration[7]), navigating relative to an internal representation of space (vector-based navigation[11,21], or orienting based on landmarks in the environment (landmark-based navigation[19]) are unified by the principle of optimal feedback control under perceptual, representational, and motor uncertainties. Simulations of navigation behavior during triangle-completion tasks reveal dynamic shifts between these strategies and their integration depending on the availability and relative uncertainty of different spatial cues. Importantly, this provides a computational-level explanation of why these seemingly different strategies are associated with different navigational biases and variability levels.

For example, the presence of landmarks reduces motor errors and, thus, endpoint variability by orienting participants in the environment[37] and continually providing visual feedback about their location[19]. Here, cue integration unfolds implicitly and dynamically but accounts only for state estimation. Computationally, landmark perception effectively corrects errors accumulated from noisy locomotor actions, thus relating path-integration to landmark-based navigation strategies[12]. The underlying internal allocentric map akin to a cognitive map also explicitly represents landmark uncertainty, which mediates how much participants should rely on landmark cues for determining their position and heading direction, which has been shown to be influential in stable vs. unstable environmental contexts[16,50,54].

The probabilistic internal representation in the context of optimal control under uncertainty provides computational-level explanations of a broad range of neurophysiological phenomena and unifies them within this conceptual framework[12]. Computationally speaking, the neural substrate underlying spatial navigation, in particular, places cells in the hippocampus[9], grid cells in the medial entorhinal cortex[55], and head-direction cells in the postsubiculum[56], serve as an internal model of the surrounding environment in relation to the actor. This internal model is sequentially updated by both internally generated self-motion cues[57] and external cues[37,58–60], supports learning and adaption in novel environments[38], and is involved in model-based planning of future actions[51,53,61].

This implies at least two other vital roles of path-integration in relation to this internal model and allows for qualitative and quantitative predictions. Namely, path-integration links sequences of egocentric observations into a unified allocentric representation, i.e., cognitive mapping[38], and determines self-location and heading based on sequences of observations of external cues and prior expectations based on the internal representation keeping track of one's current location and heading direction, i.e., reorientation[12,37]. However, due to noisy self-motion signals during path-integration, the internal representations in place, grid, and head-direction cells may not necessarily correspond to the physical location and orientation in the environment[62]. Instead, the internal representation expresses the current uncertain, subjective belief about these quantities[24], which may become increasingly discrepant from the actual state of the world over time, leading to ever-increasing spatial uncertainty and consequently navigational variability[13]. To correct for this discrepancy, the brain needs an error correction mechanism based on stable and more reliable cues, e.g., landmarks or boundary information[23,58–60] to infer position and heading relative to the internal representation[37,63].

Theoretical work suggests that place and grid cell representations may be tolerant to sensory-motor noise[13,64–66]. However, to what degree the resulting uncertainty is explicitly represented in the nervous system[67] remains unclear. Several observations and theoretical accounts suggest a potential role of uncertainty in spatial representations[12,23,68–72]. For example, the degree to which landmarks facilitate reorientation depends on their perceived stability[73,74], implying that the brain keeps track of environmental stability over time. Recent work suggests that distorted or enlarged firing patterns of place and grid cells in novel environments may code for spatial uncertainty[68,69,71], and this uncertainty is reflected in homing behavior[23].

Although the motion and observation models employed in our study are well-established in prior literature[5,45], it is important to recognize that they may not fully capture every aspect of human navigational variability observed in the homing tasks. Additionally, given our primary focus on understanding how navigators combine noisy internal self-motion cues with noisy external visual cues to update their relative position in the environment and the dynamic interaction of these sources in forming uncertain internal representations, we did not treat individual contributions of different self-motion cues separately, such as vestibular or proprioceptive information[75], and visual depth cues, including motion parallax[76]. Particularly, motion parallax is recognized as an important cue in visual place recognition models of homing behavior, where variations in scene layouts and the positioning of the viewer relative to three consistent-sized, pole-like landmarks have been shown to significantly influence the distribution of participants' homing errors[77,78].

Prior research has established that both self-motion and visual depth cues are subject to Bayesian cue integration processes[79,80]. This forms a basis for extending the current probabilistic model to support the sequential integration of multiple depth and self-motion cues according to their relative reliabilities. Additionally, other factors contributing to variability, such as individual differences, decision noise as reflected in occasional lapses from participants, and potential trial-by-trial learning effects[81], remain to be explicitly incorporated into the model's formulation. Consequently, future behavioral and modeling work should address the explicit influence of these different cues and other sources of navigational variability, specifically in the context of the triangle-completion task.

Due to the limited number of trials per participant in most datasets, we did not consider potential inter-individual variations in navigation behavior. However, prior research has shown that there are

important differences between individuals regarding their use of different strategies, which are distinguished by their navigational precision[19] or the types of errors that they make[82]. Prior research has also found that differences in navigational precision are often reflected at the population level, changing throughout development until late into life[14,83–85].

The present modeling framework provides researchers with the opportunity to generate predictions about the adoption and precision of different navigational strategies in the context of open-field navigation tasks. Specifically, it allows exploring how individuals use different strategies and how this leads to differences in navigational variability. This variability may be related to internal uncertainties of perceptual, representational, and motor processes, alongside subjective behavioral costs that balance goal achievement, cognitive map-learning, orientation, and effort minimization. Therefore, individual differences in navigation behavior and variability may have a direct correspondence to parameters in our model, making inter-individual differences quantifiable in terms of differences related to subjective uncertainties and behavioral costs. However, addressing individual differences at this level is beyond the scope of the present work and should be investigated more closely in future work.

Taken together, optimal feedback control under uncertainty provides a unifying computational account of the different sources of uncertainty from perception, internal representation, and action, their dynamic interaction in navigation, and explains the observed behavior of human participants during goal-directed navigation in both the presence and absence of landmarks. The proposed model considers the interconnected nature of navigational problems, including state estimation, locomotor control, representation learning, and path planning, while jointly accounting for uncertainty in perception, representation, and action across space and time. Importantly, the model uses state-dependent sensory uncertainty[5], signal-dependent motor variability[32], and time-dependent representation uncertainty in the internal map[33] together with recent advances in probabilistic planning[34] for efficient computation of path planning and simulation of navigation behavior.

This not only explains behavior previously deemed contradictory under cue integration models concerning the integration of landmark and self-motion cues[16,47,48] but quantitatively predicts human navigational biases and variability in a single unifying and theoretically principled framework. This provides novel opportunities for future work in neuroscience aimed at relating neuronal activity to internal perceptions instead of external stimuli or states[86,87], and in the behavioral sciences, as it makes testable predictions for behavior in spatial navigation tasks[88].

## Methods
### Experimental model and participant details
**Homing task description.** In each trial, participants viewed the environment from their starting position. They were then asked to walk an outbound path through a sequence of three-goal locations defining a three-legged outbound path. Goal locations were presented visually simultaneously from the beginning of the trial[14] or in succession after the completion of each path leg[15,16]. Once participants reached the location of the last goal, the environment turned dark. After a brief delay of 8-20 seconds, the environment turned visible again. Participants then had to walk back to the (remembered) location of the first target, i.e., homing, utilizing both internal and external cues depending on the cue condition. Internal or external cues were manipulated under one of four cue conditions. In self-motion conditions, landmarks were removed from the environment. In landmark conditions, participants were disoriented by being placed in a swivel or wheelchair, moving their position, and turning at a particular speed for a particular waiting time. In combined conditions, participants stood still for the waiting time, and landmark cues were available in the

subsequent homing response. In conflict conditions, landmarks were rotated clockwise or counterclockwise around the location of the last post unbeknownst to participants, creating a conflict between self-motion and landmark-defined home locations. Response locations were given by their stopping location, i.e., where participants believed they had successfully reached the home location, for each trial. Depending on the cue condition (self-motion, landmark, combined, conflict), different patterns of endpoint variability are observed.

**Experiment & dataset description.** We obtained endpoint data from three studies investigating the homing task with landmarks. We selected data from five experiments differing in environmental complexity within each of the studies. These experiments involved adults performing the homing task[14], comparison of the number of landmarks in the environment (Exp1a: poor vs. rich environment[16]), and adults performing the homing task in an environment with either proximal or distal landmarks[15]. The environments between the three studies differed in size, the location of targets and landmarks, and participants' starting positions at the beginning of each trial. Landmark locations remained stable across trials, while the configuration of target location was varied across trials, leading to differently shaped three-legged outbound paths (Supplementary Fig. 1). In virtual environments, i.e., the studies of Zhao & Warren[15] and Chen et al.[16], participants wore a head-mounted display that considerably restricted their field of view. The tasks between the studies further differed in the disorientation procedure within the landmark condition, i.e., speed of turning, final rotation, and positional offset, as well as the waiting time at the end of the outbound path prior to the homing response. We extracted the differing task, participant, and environmental parameters from the respective papers and the accompanying trajectory data when available (also see Supplementary Table 1).

**Ideal observer analysis (general).** Participants infer a target's location in the ideal observer account given self-motion and visual landmark cues. Using Bayes rule, the posterior distribution over target locations given visual and self-motion cues is computed by statically weighting both cues according to their relative reliability. However, the "cue reliability" is not measured perceptually but instead obtained from response variability of walking endpoints in single cue conditions. We adopt the notation by McNamara & Chen[47] to describe ideal observer analysis for the homing task. Participants infer a target's location $L$, given self-motion $S$ and visual landmark cues $V$. Using Bayes rule, the posterior distribution over target locations given visual landmark and self-motion cues is expressed as:

$$p(L|S,V) = \frac{p(S,V|L)p(L)}{p(S,V)}, \qquad (1)$$

where $P(L)$ refers to the prior distribution over possible target locations. Assuming conditional independence between self-motion and visual landmark cues, this is expressed as follows:

$$p(L|S,V) \propto p(S|L)p(V|L)p(L) \propto \frac{p(L|S)p(S)}{p_S(L)}\frac{p(L|V)p(V)}{p_V(L)}p(L). \qquad (2)$$

Assuming non-informative priors for location $L$ and both types of cues $S$ and $V$, i.e., $p(L)=p(S)=p(V)=1$, the expression further simplifies to:

$$p(L|S,V) \propto p(L|S)p(L|V). \qquad (3)$$

In the cue integration account, participants weigh self-motion cues $p(L|S)$ and landmark cues $p(L|V)$ according to their relative reliability. Cue reliability is estimated from response variability in single cue conditions by experimentally eliminating the influence of the other cue prior to the homing response, by either removing landmarks (self-

motion condition) or disorienting participants via turning (landmark condition).

**Ideal observer analysis (Euclidean distance).** To predict participants' response variability and bias in the double cue condition, we computed response variability in each cue condition as the Euclidean distance for each response to the mean response location. For the analysis, all error distributions in single cue conditions, i.e., $p(L|S)$ and $p(L|V)$, and double cue conditions, i.e., $p(L|S,V)$, are assumed to be normally distributed (self-motion: $x_{sm} \sim \mathcal{N}(\mu_{sm}, \sigma_{sm})$; landmark: $x_{lm} \sim \mathcal{N}(\mu_{lm}, \sigma_{lm})$; combined: $x_{sm+lm} \sim \mathcal{N}(\mu_{sm+lm}, \sigma_{sm+lm})$).

This assumption allows for a weighted linear combination of the two single cue Gaussian distributions relative to their response variability to predict response variability and bias in double cue conditions[14,16]. Based on the observed response variability in single cue conditions, optimal static cue weights ($w_{lm} + w_{sm} = 1$) can be computed as follows:

$$w_{lm} = \frac{1/\sigma_{lm}^2}{1/\sigma_{sm}^2 + 1/\sigma_{lm}^2} . \tag{4}$$

The variance in double cue conditions can be predicted from response variance in single cue conditions as follows:

$$\sigma_{sm+lm}^2 = w_{sm}^2 \sigma_{sm}^2 + w_{lm}^2 \sigma_{lm}^2. \tag{5}$$

This weighted average yields lower response variability in double cue conditions compared to the variance of single conditions. Further, the optimal cue weights predict participants' response location in double conditions from their response location in single cue conditions. This cue integration prediction can be explicitly tested when putting landmark cues into conflict by rotating them around the location of the last target (conflict condition). Doing so creates two response locations indicating exclusive use of either visual landmark or self-motion cues.

Proximity of participants response $\hat{r}$ to either of those location, e.g. $d_{sm} = \|\hat{\mathbf{r}} - \mathbf{x_{sm}}\|_2$. Where the self-motion consistent location $\mathbf{x_{sm}}$ is either the true target's location, e.g. (0,0), or when accounting for a particular participant's response bias by using the mean location of the self-motion consistent location[16], i.e., $\bar{x}_{sm}$. For the landmark-consistent location, the rotated normalized target location is used. It can further be corrected for participants' intrinsic response bias by subtracting the participants' mean response $\bar{x}_{lm}$ observed in the landmark condition[16]. Given both proximities to either of the single cue consistent location, i.e., $d_{sm}$ and $d_{lm}$, we calculated relative response proximity to the landmark-consistent location as follows:

$$\text{rprox}_{lm} = \frac{d_{sm}}{d_{sm} + d_{lm}} . \tag{6}$$

Response proximity to the landmark-consistent location is always between zero and one. Zero indicates participants responding at the self-motion consistent location, whereas one indicates participants responding at the landmark-consistent location. Relative response proximity can, therefore, also be interpreted as an empirical cue weight participants place on either cue, given homing responses in conflict conditions[16].

**Ideal observer analysis (heading direction).** Zhao & Warren[15] performed cue integration analysis based on heading direction rather than Euclidean distances. Heading errors were fitted to von-mises distributions for each cue condition separately (self-motion: $x_{sm} \sim \mathcal{VM}(\theta_{sm}, \kappa_{sm})$; landmark: $x_{lm} \sim \mathcal{VM}(\theta_{lm}, \kappa_{lm})$; combined: $x_{sm+lm} \sim$

$\mathcal{VM}(\theta_{sm+lm}, \kappa_{sm+lm})$). Predictions of mean heading direction and heading variability in combined cue and conflict conditions are predicted based on heading variability and heading directions from single cue conditions as follows:

$$\theta_{sm+lm} = \theta_{lm} + \Delta - \tan^{-1}(\sin(\Delta), \kappa_{lm}/\kappa_{sm} + \cos(\Delta)) \tag{7}$$

$$\kappa_{sm+lm} = \sqrt{\kappa_{sm}^2 + \kappa_{lm}^2 + 2\kappa_{sm}\kappa_{lm}\cos(\Delta)}, \tag{8}$$

where $\theta$ refers to the mean homing direction and $\kappa$ to the concentration. In conflict conditions, $\Delta$ refers to the angle of the landmark shift applied in conflict conditions. Circular standard deviations are computed based on concentration parameters $\kappa$ according to:

$$\sigma = \sqrt{2(1 - I_1(\kappa)/I_0(\kappa))}, \tag{9}$$

where $I(x)$ refers to the Bessel function of the first kind.

## Dynamic Bayesian actor model

**State space representations.** The agent state $\mathbf{x_t}$ consists of its pose $(x, y, \theta)$, including its position and heading direction, as well as the global $(x, y)$-coordinates of the landmarks (and target locations) in an allocentric frame of reference. The complete state thus can be described as $\mathbf{x} = (\mathbf{x}_{\text{pose}}, \mathbf{x}_{\text{map}})$ where $\mathbf{x}_{\text{pose}} \in \mathbb{R}^3$ and $\mathbf{x}_{\text{map}} \in \mathbb{R}^{2*L}$:

$$\mathbf{x} = \left[ \underbrace{x \quad y \quad \theta}_{\text{pose}} \quad \underbrace{m_{1;x} \quad m_{1;y} \quad \cdots \quad m_{L;x} \quad m_{L;y}}_{\text{map: landmark+goal locations}} \right]^T, \tag{10}$$

where $L$ is the number of landmark and goal locations in the environment. The current state of the world at timestep t is expressed as $\mathbf{x_t}$.

**Motion model and motor variability.** The agent changes its position and heading direction in space by sequentially performing motor actions. For the non-linear dynamics $\mathbf{x_{t+1}} = f(\mathbf{x_t}, \mathbf{u_t})$ we assume a unicycle motion model with control inputs $\mathbf{u_t} = (v_t, w_t)$. Prior research has shown that this non-holonomic motion model applies to human data for goal-oriented locomotion[31,89]. Here, we only model the discrete first-order dynamics with direct inputs of linear velocity $v_t$ and angular velocity $w_t$ for a discrete timestep $dt$:

$$\mathbf{x_{t+1}} = f(\mathbf{x_t}, \tilde{\mathbf{u}}_\mathbf{t}) = \begin{bmatrix} x \\ y \\ \theta \\ \mathbf{x}_{map} \end{bmatrix} + \begin{bmatrix} \cos(\theta) \cdot v_t \\ \sin(\theta) \cdot v_t \\ w_t \\ \mathbf{0}_{2 \cdot L \times 1} \end{bmatrix} \cdot dt . \tag{11}$$

Further, we assume that these inputs $\mathbf{u_t} = (v_t, w_t)$ are subject to signal-dependent noise[32] $\boldsymbol{\alpha}$ with noise parameters ($\alpha_1, \alpha_2, \alpha_3, \alpha_4$):

$$\tilde{\mathbf{u}}_\mathbf{t} = \mathbf{u_t} + \mathcal{N}(\mathbf{0}, \mathbf{Q}_t) \text{ with}$$
$$\mathbf{Q}_t = Q(\mathbf{u_t}, \boldsymbol{\alpha}) = \begin{pmatrix} \alpha_1^2 v^2 + \alpha_2^2 w^2 & 0 \\ 0 & \alpha_3^2 v^2 + \alpha_4^2 w^2 \end{pmatrix} \tag{12}$$

**Observation model and observation variability.** At each timestep, the agent makes noisy egocentric observations $\mathbf{z_t}$, i.e., relative distance and bearing[17], of landmarks and objects in the environment. When the agent encounters landmarks or objects for the first time, observations are converted to allocentric coordinates and then stored within the internal representation[64]. The non-linear observation model $\mathbf{z_t} = h(\mathbf{x_t}) + \mathbf{r_t}$ consists of observations with additive Gaussian noise. Once in the field of view of the agent, an observation of a landmark l consists of its distance and direction relative to the agent, which can be

expressed as:

$$\mathbf{z_{t;l}} = h(\mathbf{x_t}) = \begin{bmatrix} \sqrt{(x_{t;x} - m_{t;l;x})^2 + (x_{t;y} - m_{t;l;y})^2} \\ \tan^{-1}\left(\frac{x_{t;y} - m_{t;l;y}}{x_{t;x} - m_{t;l;x}}\right) \end{bmatrix}. \qquad (13)$$

We assume that correspondence between a measurement and landmark l is unambiguously known. The agent observes landmarks or targets if they fall inside the field of view of the agent. Thus the complete environment is not observable to the agent at once. Further, observations are subject to state-dependent Gaussian noise, which depends on the current position and heading direction of the agent relative to the observed object, with noise parameters $\boldsymbol{\epsilon} = (\sigma_{r_{min}}, \sigma_{r_{max}}, \sigma_{\psi_{min}}, \sigma_{\psi_{max}})$.

Observations of an object's distance get noisier with increasing distance and the more peripheral they are relative to the center of the agent's field of view:

$$\sigma_r = d_l \cdot \left(\sigma_{r_{max}}^2 (1 - \cos(\beta)) + \sigma_{r_{min}}^2\right), \qquad (14)$$

where $d_l$ refers to the distance and $\beta_l$ refers to the angle between the agent's gaze direction and the vector between the agent and the landmark or target. Similarly, noise in the perception of the bearing of an object relative to the observer increases with both distance and relative to the center of the field of view[5,25]:

$$\sigma_\psi = \log(d_l) \cdot \left(\sigma_{\phi_{max}}^2 (1 - \cos(\beta)) + \sigma_{\phi_{min}}^2\right). \qquad (15)$$

Thus the perceptual uncertainty in the current observation of a single landmark can be expressed as follows:

$$\tilde{\mathbf{z}}_t = \mathbf{z_t} + \mathcal{N}(\mathbf{0}, \mathbf{R}_t) \text{ with}$$
$$\mathbf{R_t} = R(\mathbf{x_t}, \boldsymbol{\epsilon}) = \begin{pmatrix} \sigma_r & 0 \\ 0 & \sigma_\psi \end{pmatrix}. \qquad (16)$$

**Belief-space representation.** As the true state of the world can only be perceived indirectly, the agent only has access to the belief state $b_t = (\boldsymbol{\mu_t}, \boldsymbol{\Sigma_t})$. This extends the state-space formulation from Eq. (10) by an explicit representation of spatial uncertainty over the agent's position, heading, landmark, and goal locations.

The belief $\mathbf{b_t}$ at timestep $t$ describes this as follows ($t$ dropped for notational convenience):

$$\boldsymbol{\mu_t} = \begin{pmatrix} \boldsymbol{\mu_{pose}} \\ \boldsymbol{\mu_{map}} \end{pmatrix} \qquad \boldsymbol{\Sigma_t} = \begin{pmatrix} \boldsymbol{\Sigma_{pose}} & \boldsymbol{\Sigma_{pose-map}} \\ \boldsymbol{\Sigma_{map-pose}} & \boldsymbol{\Sigma_{map}} \end{pmatrix}, \qquad (17)$$

with the following submatrices which express the uncertainty about the current position and heading, the internal map, and the relation between those:

$$\boldsymbol{\Sigma_{pose}} = \begin{pmatrix} \sigma_{xx} & \sigma_{xy} & \sigma_{x\theta} \\ \sigma_{yx} & \sigma_{yy} & \sigma_{y\theta} \\ \sigma_{\theta x} & \sigma_{\theta y} & \sigma_{\theta\theta} \end{pmatrix},$$

$$\boldsymbol{\Sigma_{map}} = \begin{pmatrix} \sigma_{m_{1;x}m_{1;x}} & \sigma_{m_{1;x}m_{1;y}} & \cdots & \sigma_{m_{1;x}m_{n;x}} & \sigma_{m_{1;x}m_{n;y}} \\ \sigma_{m_{1;y}m_{1;x}} & \sigma_{m_{1;y}m_{1;y}} & \cdots & \sigma_{m_{1;y}m_{n;x}} & \sigma_{m_{1;y}m_{n;y}} \\ \vdots & \vdots & \ddots & \vdots & \vdots \\ \sigma_{m_{n;x}m_{1;x}} & \sigma_{m_{n;x}m_{1;y}} & \cdots & \sigma_{m_{n;x}m_{n;x}} & \sigma_{m_{n;x}m_{n;y}} \\ \sigma_{m_{n;y}m_{1;x}} & \sigma_{m_{n;y}m_{1;y}} & \cdots & \sigma_{m_{n;y}m_{n;x}} & \sigma_{m_{n;y}m_{n;y}} \end{pmatrix},$$

$$\boldsymbol{\Sigma_{pose-map}} = \begin{pmatrix} \sigma_{xm_{1;x}} & \sigma_{xm_{1;y}} & \cdots & \sigma_{xm_{n;x}} & \sigma_{xm_{n;y}} \\ \sigma_{ym_{1;x}} & \sigma_{ym_{1;y}} & \cdots & \sigma_{ym_{n;x}} & \sigma_{ym_{n;y}} \\ \sigma_{\theta m_{1;x}} & \sigma_{\theta m_{1;y}} & \cdots & \sigma_{\theta m_{n;x}} & \sigma_{\theta m_{n;y}} \end{pmatrix}, \qquad (18)$$

$$\boldsymbol{\Sigma_{map-pose}} = \begin{pmatrix} \sigma_{m_{1;x}x} & \sigma_{m_{1;x}y} & \sigma_{m_{1;x}\theta} \\ \sigma_{m_{1;y}x} & \sigma_{m_{1;y}y} & \sigma_{m_{1;y}\theta} \\ \vdots & \vdots & \vdots \\ \sigma_{m_{n;x}x} & \sigma_{m_{n;x}y} & \sigma_{m_{n;x}\theta} \\ \sigma_{m_{n;y}x} & \sigma_{m_{n;y}y} & \sigma_{m_{n;y}\theta} \end{pmatrix}.$$

Having access to the full belief state, rather than a single point estimate of self and landmark locations allows for the model to keep track of the growing positional uncertainty and resolve it later by actively reorienting and thereby combining noisy observations of landmarks with prior expectations based on the internal map about their relative position.

**Bayesian filtering: belief dynamics & state estimation.** The framework of Bayesian filtering provides a principled manner in which the belief state is updated as a consequence of perception and action[12,24,90]. Central to these computations are the two steps of prediction and correction. The prediction step uses a probabilistic motion model of the form $p(\mathbf{x_t}|\mathbf{x_{t-1}}, \mathbf{u_t})$ and increases positional uncertainty. The correction step uses an observation model of the form $p(\mathbf{z_t}|\mathbf{x_t})$, which incorporates noisy landmark observations from the environment with prior uncertain expectations about the landmarks' locations stored in the internal representation. Both steps can be performed by recursively applying the probabilistic Bayes Filter to obtain the posterior distribution over possible states $p(\mathbf{x_t}|\mathbf{z_{1:t}}, \mathbf{u_{1:t}})$ given actions and observations as follows:

$$\eta \cdot \underbrace{p(\mathbf{z_t}|\mathbf{x_t})}_{\text{correction step}} \cdot \int \underbrace{p(\mathbf{x_t}|\mathbf{x_{t-1}}, \mathbf{u_t}) \cdot p(\mathbf{x_{t-1}}|\mathbf{z_{1:t-1}}, \mathbf{u_{1:t-1}}) \, dx_{t-1}}_{\text{prediction step}}. \qquad (19)$$

Choosing functions the non-linear dynamics and observation functions f and h such that they are differentiable allows for local linearization around the current state and application of an Extended Kalman Filter[4,23,90,91]. This allows for a closed-form belief propagation of the form $(\boldsymbol{\mu_t}, \boldsymbol{\Sigma_t}, \mathbf{u_t}, \mathbf{z_t}) \rightarrow (\boldsymbol{\mu_{t+1}}, \boldsymbol{\Sigma_{t+1}})$.

The equations for the prediction step then become:

$$\bar{\boldsymbol{\mu}}_{t+1} = f(\boldsymbol{\mu_t}, \mathbf{u_t})$$
$$\bar{\boldsymbol{\Sigma}}_{t+1} = \mathbf{A_t} \boldsymbol{\Sigma_{t-1}} \mathbf{A_t}^T + \mathbf{G_t} \mathbf{Q_t} \mathbf{G_t}^T, \qquad (20)$$

where

$$\mathbf{A_t} = \frac{\partial f(\boldsymbol{\mu_t}, \mathbf{u_t})}{\partial \mathbf{x_t}}$$
$$\mathbf{G_t} = \frac{\partial f(\boldsymbol{\mu_t}, \mathbf{u_t})}{\partial \mathbf{u_{t-1}}}. \qquad (21)$$

Further, representation uncertainty $\mathbf{V_t}$ gets added to landmarks $m_L := (m_{L;x}, m_{L;y})$ at each timestep

$$\boldsymbol{\Sigma_{map}}_{m_{L;t+1}} = \boldsymbol{\Sigma_{map}}_{m_{L;t}} + \mathbf{V_t}, \qquad (22)$$

where

$$\mathbf{V_t} = \begin{pmatrix} \delta_{xy} & 0 \\ 0 & \delta_{xy} \end{pmatrix}. \qquad (23)$$

The equations for the correction step then become:

$$\mathbf{K}_t = \bar{\Sigma}_{t+1}\mathbf{H}_t^T(\mathbf{H}_t\bar{\Sigma}_{t+1}\mathbf{H}_t^T + \mathbf{R}_t)^{-1}$$
$$\boldsymbol{\mu}_{t+1} = \bar{\boldsymbol{\mu}}_{t+1} - \mathbf{K}_t(h(\bar{\boldsymbol{\mu}}_{t+1}) - \mathbf{z}_t) \qquad (24)$$
$$\Sigma_{t+1} = (\mathbf{I} - \mathbf{K}_t\mathbf{H}_t)\bar{\Sigma}_{t+1},$$

where

$$\mathbf{H}_t = \frac{\partial h(\boldsymbol{\mu}_t, \mathbf{u}_t)}{\partial \mathbf{x}_t}. \qquad (25)$$

**Representation variability.** We assumed that the agent's internal representation is noisy, such that precise memory of objects and landmark locations decays over time[33]. Internally represented landmark locations $m_{L;t} := (m_{L;x}, m_{L;y})$ follow a 2D random walk[92] according to:

$$\mathbf{m}_{\mathbf{L};t+1} = \mathbf{m}_{\mathbf{L};t} + N(\mathbf{0}, \mathbf{V}_t). \qquad (26)$$

**Belief-space planning.** The agent plans motor actions based on the current belief about its pose and the location of landmarks within the environment $(\boldsymbol{\mu}_t, \Sigma_t)$ by propagating it according to the belief dynamics, minimizing the cost-function $J$[34,91,93]. The planning step results in a sequence of belief states $\boldsymbol{\mu}_{0:T}$ and planned actions $\boldsymbol{u}_{0:T-1}$ and can be written as a constrained non-linear trajectory optimization problem of the form:

$$\min_{\boldsymbol{\mu}_{0:T}, \boldsymbol{u}_{0:T-1}} \quad J(\boldsymbol{\mu}_{0:T}, \Sigma_0, \boldsymbol{u}_{0:T-1}) \qquad (27a)$$

$$\text{subject to} \quad \boldsymbol{\mu}_{t+1} = f(\boldsymbol{\mu}_t, \mathbf{u}_t), \qquad (27b)$$

$$\boldsymbol{\mu}_t \in X_{\text{feasible}}, \qquad (27c)$$

$$\boldsymbol{u}_t \in U_{\text{feasible}}, \qquad (27d)$$

where T refers to the planning horizon and J to the cost-function. Feasible values for the optimization are those which satisfy the following lower and upper bounds:

$$X_{\text{feasible}} \begin{cases} x \in [x_{min}, x_{max}] \\ y \in [y_{min}, y_{max}] \\ \theta \in [\theta_{min}, \theta_{max}] \\ \mathbf{m}_{1:L;\mathbf{x}} \in [m_{x_{min}}, m_{x_{max}}] \\ \mathbf{m}_{1:L;\mathbf{y}} \in [m_{y_{min}}, m_{y_{max}}] \end{cases} \quad U_{\text{feasible}} \begin{cases} v \in [v_{min}, v_{max}] \\ w \in [w_{min}, w_{max}] \end{cases}$$

$$(28)$$

We only include the mean states $\boldsymbol{\mu}_t$ as well as the control inputs $\mathbf{u}_t$ in our optimization problem formulation. Excluding the covariance from the optimization variables reduces the complexity of the planning problem, thus speeding up runtime[34]. This allows for effective re-planning at every timestep, which is necessary to incorporate closed-loop feedback into the system via model predictive control. In order to deal with future observations, yet unknown to the agent during planning, we assume that any stochasticity is due to Gaussian noise, so the agent uses maximum likelihood observations during planning, i.e., $\mathbf{z}_t = h(\boldsymbol{\mu}_t)$[93]. Under this set of assumptions, the problem of belief-space planning becomes open loop and deterministic.

The restricted field-of-view of the agent leads to partial observability of the environment, which needs to be considered by the agent during planning to select appropriate actions that reduce uncertainty about task-relevant quantities. In order to include landmarks currently not visible in the planning of future actions, we change Eq. (24) for the

belief-space planning problem[91]. More specifically, we include $\Delta_{t+1}$, a diagonal matrix comprising of a vector $\delta_{t+1}$, which indicates which landmarks are currently observable to the agent and which are not (maximum likelihood observations). Rather than using a strict indicator function to determine the visibility of a landmark, which complicates optimization due to discontinuity, its visibility is approximated continuously in an iterative fashion using a sigmoid function, i.e., $1 - \frac{1}{1+\exp^{(-a^*x)}}$, for decreasing values for $a$, together with signed distances $x$ to each object and landmark relative to the agent's current belief about the position and heading. This changes the computation of $\mathbf{K}_t$ for the planning belief dynamics in the update step as follows:

$$\mathbf{K}_t = \bar{\Sigma}_{t+1}\mathbf{H}_t^T\Delta_{t+1}(\Delta_{t+1}\mathbf{H}_t\bar{\Sigma}_{t+1}\mathbf{H}_t^T + \mathbf{R}_t)^{-1}\Delta_{t+1}. \qquad (29)$$

**Cost-function.** Which trajectory the agent plans and executes is strongly influenced by the choice of the cost-function. A cost-function for navigation should entail reaching a target in space

$$J_{\text{goal}}(\boldsymbol{\mu}_t) = \left(\boldsymbol{\mu}_{pose_{t;(x,y)}} - \boldsymbol{\mu}_{map_{g_t}}\right)^T \mathbf{C}_\mathbf{q} \left(\boldsymbol{\mu}_{pose_{t;(x,y)}} - \boldsymbol{\mu}_{map_{g_t}}\right), \qquad (30)$$

reducing the uncertainty about self-location,

$$J_{\text{position-uncertainty}}(\Sigma_t) = tr\left(\Sigma_{t;\text{pose}}\right), \qquad (31)$$

and the location of goals and landmarks,

$$J_{\text{map-uncertainty}}(\Sigma_t) = tr\left(\Sigma_{t;\text{map}}\right), \qquad (32)$$

while penalizing control effort

$$J_{\text{control}}(\mathbf{u}_t) = \mathbf{u}_t^T\mathbf{C}_\mathbf{r}\mathbf{u}_t + C_{tr}\left(v_t \cdot w_t\right)^2. \qquad (33)$$

Given a set of feasible states and controls $\boldsymbol{\mu}_t$ and an internal belief about the location of the goal $\mathbf{g}_t = (m_{\mu_{L;x,t}}, m_{\mu_{L;y,t}})$, the objective function $J$ can be computed by propagating the initial belief state $\mathbf{b}_0 = (\boldsymbol{\mu}_0, \Sigma_0)$ with the sequence of controls $\boldsymbol{u}_{0:T}$ using Equations (20) and (24):

This leads to the following weighted cost-function for the planning problem with planning horizon $N$, $J_{\text{cost}}(\mu_t, t+N, \Sigma_{t,t+N}, u_{t,t+N}) =$

$$w_1 \sum_{t=0}^{n} J_{goal}(\boldsymbol{\mu}_t) + w_2 \sum_{t=0}^{n} J_{\text{position-uncertainty}}(\Sigma_t)$$
$$+ w_3 \sum_{t=0}^{n} J_{\text{map-uncertainty}}(\Sigma_t) + w_4 \sum_{t=0}^{n} + J_{\text{control}}(\mathbf{u}_t), \qquad (34)$$

with weights $\mathbf{w} = (w_1, w_2, w_3, w_4)$. This set of weights captures the general trade-off between sensing, i.e., remaining oriented by reducing uncertainty about agent position and landmark position and acting, i.e., reaching a goal. Once the trajectory is planned for $T$ steps (planning horizon), the agent executes the first $N$ steps of the current plan (control horizon) and then replans the trajectory based on an updated internal belief.

**Reorientation.** Due to the nature of the task and noise in the motor system, the agent's internal model and the true state of the world could become so divergent that the agent's planning procedure may get stuck. For example, suppose the agent is completely disoriented. In that case, e.g., it expects to see landmarks based on the belief about its own heading direction and its internal belief about the landmarks locations but does not observe any landmarks. In this case, the agent will keep turning in the direction it believes landmarks are until it brings them into the field of view. The observation is then integrated with the internal model via Equation (24), providing reorientation so

that the agent can further plan and act accordingly to its subjective beliefs about the environment.

## Model simulations & data analysis

**Simulation of homing task conditions.** For the homing task, the simulation of the agent navigating the outbound path was the same for all four homing conditions. Targets were either visible to the agent simultaneously from the beginning of the trial or appeared in succession during the unfolding of the triangle-completion task. Both the location of landmarks and targets were initialized at the location where the virtual agent first encountered them. Each trial was simulated identically and independently, i.e., with no trial-by-trial learning. For the inbound path, the agent had to wait 8–20 seconds before homing, depending on the experiment. We increased the agents' uncertainty about landmark locations by applying a small amount of process noise to simulate the effect of time-dependent representation variability (Equation (22)) during the waiting and disorientation periods respectively. We further modeled the different cue conditions during the homing response by manipulating the reliability or availability of the different cues. In self-motion conditions, we removed the availability of landmark cues during the homing path. In landmark conditions, we disoriented the agent by simulating a rotating swivel chair. For this, we first applied a slight positional offset to the agent's position and the internal belief, which we extracted from trajectory data (if available) or assumed a Gaussian distribution similar in magnitude. After this, we simulated the agent's disorientation by rotating them in their location using the turning speeds of the original studies, i.e., updating the heading direction using noisy velocity inputs, and by increasing the overall uncertainty in the agent's belief regarding heading and positional estimates. In combined conditions, we had the agent wait for n seconds before homing. Similarly, the agent remained still in conflict conditions for n seconds, and landmarks were rotated 15-90 degrees, depending on the experiment, around the location of the third post before homing. The agent then performed navigation towards the location of the goal utilizing the belief and available landmarks to regain orientation and correct for errors due to noisy actions.

**Model simulation.** For each of the experiments, we obtained the number of trials for each of the four conditions, the participants starting position and sequence of goal locations, and other relevant differences between environments and task procedures (Supplementary Table S2). The optimal control formulation (Equation (27)) was implemented in the CasADI framework[94], which allows for the specification of complex cost-functions and automatic differentiation. Planning is achieved by repeatedly solving a non-linear program using IPOPT and the HSL solvers (academic license). Trials were executed in parallel on a compute cluster using Ray Tune[95], which allowed for efficient scheduling and logging of individual trial results. Each trial, due to the complexity of belief-space planning involving continuous states, non-linear actions, and observations, takes 5-10 minutes on a single CPU core. Simulation of all experiments (5672 trials) takes between 120-180 minutes across 13 compute nodes, when running 96 trials on each node concurrently.

**Model parameters.** Given the high computational demands of model simulations, it was not possible to fit the model parameters to empirically observed endpoint data using numerical optimization methods. Instead, we used a combination of constraining parameters on the basis of previous research, e.g., the motor variability, or extracting parameter values from trajectory data, e.g., the average speed of walking, and finally by inspecting the endpoint distributions obtained from model simulations. Initial values for all model parameters were chosen based on previous literature and sequentially adjusted manually so as to match the endpoint distributions in the Nardini et al.[14] study with the exception of the cost-function parameters (see Supplementary Table 2 for parameter values). The reason is, that the Nardini et al.[14] data does not contain trajectory data but only endpoint distributions. The cost-function parameters are known to strongly influence the shape of the trajectories[45,46]. In order to set the cost-function parameters initially, we therefore adjusted the cost-function parameters in such a way that the model's walking trajectories resembled those from available data, i.e., Chen et al.[16] and Zhao & Warren[48]. Note, that our additional analysis in the section confirms that endpoint distributions did not change dramatically over a parameter range between 0 and 10 for each parameter when drawing random parameter values for the Nardini et al.'s study[14].

Similarly, the planning and control horizons did not change endpoint distributions drastically over a range of 0.5 to 3.0 seconds for the planning horizon and 0.5 to 1.5 seconds for the control horizon. Consequently, the planning and control horizons were set primarily due to practical constraints on computation to make the simulations of many trials, experiments, and the different models in the context of model ablations and model comparisons feasible. Motion limit parameters were chosen based on established literature on human locomotion[96] in conjunction with extracted values from trajectory data by Zhao & Warren[50] and Chen et al.[16]. Initial belief state and initial uncertainty were set based on the assumption that participants did not know about the layout of the environment at the beginning of a trial. The initial positional uncertainty was extracted from trajectory data from the variance of starting locations.

Importantly, to test whether the navigation model generalizes to navigation data from multiple other experimental settings and conditions across different labs, we used a single set of parameters for all simulations instead of adjusting parameters to each experiment or condition separately. This can be considered a form of transfer testing and the above results validate the model's ability to capture navigation data across triangle-completion tasks. In the following, we provide more detail about noise parameter settings individually. Importantly, we assumed that these parameters are equal for both the generative model of the task and the participants' internal model.

The observation noise parameters are responsible for errors the actor makes when observing landmarks at different distances and directions relative to the current viewing direction. They further shape the magnitude of observation uncertainty in the correction step, affecting how much positional information the actor can gain from a given landmark as well as shaping the uncertainty of landmarks in the internal map. Importantly, observation noise is based on the fact that integrated depth cue reliability and distance estimation accuracy diminish with increased distance, as evidenced in the majority of depth cues[97–100], and the parameters also reflect increased noise in peripheral vision due to decreased visual acuity[101,102]. The magnitude of these four parameters regarding the observation noise was constrained based on previous modeling work using similar observation models of landmarks/objects, in which the magnitude of the noise scales with the distance and direction away from the observer[5,25].

The motor noise parameters are responsible for signal-dependent errors in response to linear and angular velocity changes and directly shape the actor's internal positional and heading uncertainties via the prediction step. These capture any variability that is associated with the execution of motor actions, including changes of position through linear forward movement and changes of heading direction through turning of the body. The paper by Kallie et al.[89] provides basic estimates of step length variability (linear-linear velocity noise) and veering variability (linear-angular velocity noise) from a locomotor task (see Figure 8 and Table 5/6 of the paper), which we converted from step-based to velocity-based units. Both parameters were further adjusted to fit the velocity profiles and distributions of velocity present in the trajectory data from Chen et al.[16] and Zhao & Warren[15]. Variability in the linear component from turning (angular-linear velocity noise), i.e., forward movement when turning on the spot, was set to be

minimal because the homing task did not necessarily require large turns; their impact can therefore be considered negligible. Variability in turning (angular-angular velocity noise) was constrained in its overall magnitude using the paper by Jürgens et al.[103] (active turning condition AT).

The representation noise parameters influence the agent's uncertainty regarding internal estimates of landmark and object positions, which decay over time. Although the real-world dynamics for the navigation task do not include the movement of objects or landmarks, we introduced this parameter to model the fact that agents cannot perfectly remember the position of landmarks and objects that they have previously encountered[33]. Therefore, the representation noise is best thought of as a process noise parameter, increasing the uncertainty of the internal map representation over time. Given constraints on the other two parameters, we freely selected this parameter, assuming the variability would be equal in the $x$ and $y$ directions. Importantly, this representation noise remained constant throughout the experiment because all trials were simulated identically and independently from each other, and therefore, does not account for any trial-by-trial variability, e.g., due to learning.

**Model comparison & model ablation.** To further understand the complex interaction of the three sources of sensorimotor noise, we compared endpoint variability from different versions of the dynamic Bayesian actor model to empirical data by selectively excluding particular sources of variability. These models either considered variability and therefore uncertainty in perception, representation, and action jointly (full model), separately (perceptual variability only; motor variability only; representation variability only), without one of the sources (perceptual variability & representation variability; representation & motor variability; motor & perceptual variability), or not at all (zero model). Each of the eight different models was simulated for each experiment 10 times. For each endpoint distribution within the four conditions, we computed the energy distance[104] between the observed distributions in the experiment and the simulated endpoint distributions from each model. This allowed us to assess how well each of the different models can account for the endpoint variability observed in the respective experiments and how much and in what manner the three sources of variability/uncertainty contributed to the observed endpoint variability.

**Data preprocessing & replication of prior analysis.** All endpoint data, i.e., empirical data from the respective studies and data from model simulations, were reanalyzed following the procedures used in prior work[14–16]. We normalized homing responses for each homing location by centering them within a common coordinate system, with either the actual homing direction as the north-direction[15] or centered around the true target location[14,16]. Outliers were removed based on distances to the mean location within each condition by excluding those trials that exceeded the third quartile by three times the interquartile range[14,16]. For data of Chen et al.[16] and Zhao & Warren[15], we also removed biases in the four individual target locations before pooling them together so as not to increase response variability artificially[16]. After pooling data from all four target locations, we computed the response variability for each of the simulated and real participants in each of the four conditions. We did so either as the standard deviation of the Euclidean distances (or heading directions) between the response locations for each trial or the mean response location (or heading) across all trials. We used computed response variances for the two single cue conditions (self-motion and landmark) to predict response variance for the combined cue condition according to the perceptual cue integration model for Euclidean distance[14,16] or heading direction[15]. We considered the integration of cues statistically optimal when the reduction in response variability in the combined cue conditions can be predicted from response variability in single cue

conditions (Equation (5)). Further, we used mean responses in conflict conditions to predict relative response proximity to the landmark or self-motion-defined target locations. We compared them with optimal predictions based on response variability. All data were analyzed at the single participant level if not indicated otherwise.

**Statistical tests.** We tested endpoint distributions for the different cue conditions for multivariate normality (Henze Zirkler test), but this assumption was violated for most endpoint distributions. Therefore, we resorted to using energy statistics[104] from the python package dcor (version 0.6), which allows for quantifying the degree of similarity (via energy distance) and testing for equality (via homogeneity energy test) of random vectors sampled from arbitrary multi-dimensional non-parametric distributions. For the processed response error and variability data, we utilized the Shapiro-Wilk test to check for normality. Having confirmed that the data were normally distributed, we then conducted a two-way repeated measures ANOVA on response variability, with the type (model simulation vs. empirical data) and cue condition (self-motion, landmark, combined, conflict) as the independent variables. Following ANOVAs, we performed post-hoc tests using pairwise $t$-tests (two-sided). All tests were performed using the Python package pingouin[105] (0.5.1). We considered results significant if $p < 0.05$. Multiple comparisons were corrected using Bonferroni corrections.

### Reporting summary
Further information on research design is available in the Nature Portfolio Reporting Summary linked to this article.

## Data availability
The model data generated in this study has been deposited on *Zenodo*. This work re-analyses data from previous studies, namely, Nardini et al. (2008), Chen et al. (2017), and Zhao & Warren (2015). The data is available upon request from the authors of the original studies. Source data are provided in this paper.

## Code availability
The model code and custom analysis scripts used in this paper have been deposited on *Zenodo*.

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

## Acknowledgements

We thank Marko Nardini, Xiaoli Chen, Timothy P. McNamara, Mintao Zhao, and William H. Warren for providing their behavioral data and further clarification regarding the experimental and analysis procedures. Calculations for this research were conducted on the Lichtenberg high-performance computer of the TU Darmstadt. This research was supported by the European Research Council (ERC; Consolidator Award "ACTOR"-project number ERC-CoG-101045783 to F.K. and C.R.) and "The Adaptive Mind," funded by the Excellence Program of the Hessian Ministry of Higher Education, Research, Science and the Arts (to C.R.).

## Author contributions

Conceptualization, F.K., J.F., and C.R.; Methodology, F.K. and C.R.; Investigation, F.K. and C.R.; Visualization, F.K. and C.R.; Formal analysis, F.K and C.R.; Validation, F.K. and C.R.; Writing—original draft, F.K.; Writing—review & editing, F.K, J.F., and C.R., Funding acquisition: J.F. and C.R.; Supervision: J.F. and C.R.

## Funding

## Competing interests

The authors declare no competing interests.
