## [Peer Review File · Nature Communications]

Human navigation strategies and their errors result from dynamic interactions of spatial uncertaintiesREVIEWER COMMENTS

Reviewer #1 (Remarks to the Author):

In this well-written study the authors propose a dynamic Bayesian actor model of navigation under the framework of optimal feedback control, which takes into account the sequential and dynamic interaction of uncertainties in perception, internal representations, and actions. This model also accounts for state estimation, learning, path-planning, and control while acknowledging the state-dependent, signal-dependent, and time-dependent uncertainties.

The research findings provide a unifying account of goal-directed navigation, showing how different behaviors and strategies emerge from the interaction of uncertainties under various environmental conditions. The model can predict errors and variability of different navigation behaviors across multiple experiments and sheds light on some contradictory phenomena reported in previous work.

The study concludes that perception, cognition, and action in spatial navigation are inseparably intertwined and should be considered jointly, emphasizing the fundamental role of the dynamic evolution of uncertainties during goal-directed navigation.

I found the emphasis on an integrated, holistic perspective on the cognitive and perceptual dynamics of the action-perception cycle in navigation to be very timely and a much needed contribution, because much of the field still focuses on individual aspects of this process in isolation and thereby runs the risk of missing the level of description that best captures the essence of the cognitive mechanisms at play during spatial navigation.

Overall most of the claims of the study seem to be in alignment with underlying data. However, I do have the following comments:

Major:

- Navigation strategies. individual differences in navigation strategies need to be discussed.
- An important cue in navigation is motion parallax, as it can provide both depth and angular information and may be used for Euclidean triangulation. It seems that motion parallax information was present in the triangle completion experiments. Why was this not included in the model? At minimum I feel this source of information needs to be discussed and whether it would affect the results or not. In particular the observation model/ observation variability (page 11, line 849-858) presumes that object or landmark information becomes more noisy with larger distances, and at more peripheral angles of the visual field. During forward motion the opposite is true in some sense for motion parallax. A landmark that is infinitely far away provides the most reliable background for parallax information with an intermediate landmark/ object, and the angular change of the parallax cue for a given distance traveled is maximal towards the periphery of the visual field (max at 90° to the traveling direction). This seems to be at odds with the assumptions in the observation model.
- It is laudable that the authors explicitly examine the role of different sources of uncertainty. It would be helpful if more numerical reporting of the different types of uncertainties were reported, in given context and across contexts. For example, at which relative uncertainty levels of sensory, motor, and representational noise do simulated or biological agents switch strategies? How do the different types of uncertainty compare to each other in magnitude/ impact?
- Page 6 line 369. It is stated that the results allow to 'causally attribute the observed behavior' to the different, dynamically changing sources of uncertainty. What exactly do the authors base this on? Fitting the outcome of the 8 models to the experimental data does, in my opinion, not warrant any causal claims. Correlation is not causation. Also see page 9, line 578. A related point is that the manuscript does not discuss caveats adequately, for example the fact that the choices of models and parameters are necessarily limited. There may be, and probably are, model designs that would capture the behavior and data even better. This does not take away from the contribution of the current work, demonstrating how different sources of uncertainty can dynamically interact. However, statements like 'This provides invaluable insights..' seem a bit exaggerated to me.

- It seems that the authors suggest that their model(s) can fully explain the origin of different navigation strategies. However, there is ample evidence that individual differences (across people) in navigation strategies are persistent to some degree, and are linked to different neural systems and brain regions (see for example Iaria et al. 2003 Cognitive Strategies Dependent on the Hippocampus and Caudate Nucleus in Human Navigation: Variability and Change with Practice, or Chersi & Burgess 2015 The cognitive architecture of spatial navigation: hippocampal and striatal contributions). I feel the ample literature on individual differences in brain, behavior, and navigation strategy needs to be discussed, and what the implications are for the current work.

Minor:

- Please consider making the simulation scripts openly accessible.
- Figure 1d. Information missing on what the red and blue cross in the 'Proximity' panel represent (the mean?).
- Figure 3 panel a. Information missing on what the red crosses represent. Also, there is a visibility issue with the red cross on top of red circles in the self-motion scatter plots.
- page 22, line 1271: '..targets location' typo
- page 22, line 1273 '..targets locations' typo again?

Reviewer #2 (Remarks to the Author):

The manuscript presents several lines of results to support the bold hypothesis that patterns of endpoint bias and variability across multiple triangle completion experiments can be explained with a single dynamic Bayesian actor model of navigation.

The findings would be very interesting and noteworthy if the model fitting and comparison procedures were sound, but I have a few questions before I am reassured:

1) Line 1251 states that the parameters were "manually fitted". This seems to beg the question of how the 28 parameters listed in Table 2 could be optimized effectively.

1a) It is mentioned that "noise parameters were constrained by findings from previous experiments", but to me it was unclear whether this encompasses all or only some of motor, observation, and representation noise parameters. Clarifying exactly which ones were constrained by which experiment would be helpful.

1b) Even if all three kinds of noise parameters were constrained by previous experiments, it would still leave 19 parameters to optimize. It would be more reassuring if it was shown that either the exact value of the parameters didn't matter (e.g., by performing grid search and showing the landscape of the loss function), or that the values were reasonable a priori.

For example, I wonder whether different values for the cost-function weights and/or the noise parameters not constrained by previous experiments (if any) would change the results qualitatively and make the other models (e.g., those without representation or motor noise) perform better.

2) Related to 1), I wonder whether parameters were optimized again when considering the cases when one or more of the motor, observation, or representation noise were ignored (Fig 3h). If we optimize the parameters for the case when none of them are ignored, we can certainly expect the model to perform worse when some of them are ignored, due to overfitting.

3) It is unclear which results were used to fit the parameters and which results were predicted. In Line 269, it sounds like the parameters were fit to the results from Nardini et al. 2008 and were used to predict results from Zhao et al. 2015 and Chen et al. 2015. However, in Line 1251, it is stated that the parameters were "fitted to trajectory and endpoint data for "all experiments simultaneously". Which

one was the case?

Minor point

4) Units need to be specified for parameters in Table 2.

Reviewer #1 (Remarks to the Author):

In this well-written study the authors propose a dynamic Bayesian actor model of navigation under the framework of optimal feedback control, which takes into account the sequential and dynamic interaction of uncertainties in perception, internal representations, and actions. This model also accounts for state estimation, learning, path-planning, and control while acknowledging the state-dependent, signal-dependent, and time-dependent uncertainties.

The research findings provide a unifying account of goal-directed navigation, showing how different behaviors and strategies emerge from the interaction of uncertainties under various environmental conditions. The model can predict errors and variability of different navigation behaviors across multiple experiments and sheds light on some contradictory phenomena reported in previous work.

The study concludes that perception, cognition, and action in spatial navigation are inseparably intertwined and should be considered jointly, emphasizing the fundamental role of the dynamic evolution of uncertainties during goal-directed navigation.

I found the emphasis on an integrated, holistic perspective on the cognitive and perceptual dynamics of the action-perception cycle in navigation to be very timely and a much needed contribution, because much of the field still focuses on individual aspects of this process in isolation and thereby runs the risk of missing the level of description that best captures the essence of the cognitive mechanisms at play during spatial navigation.

Response #1:

We would like to thank the reviewer very much for this statement that we fully agree with!

Overall most of the claims of the study seem to be in alignment with underlying data. However, I do have the following comments:

Major:

- Navigation strategies. Individual differences in navigation strategies need to be discussed.

Response #2:

We thank the reviewer for raising this point. It appears to be connected to the reviewer's last major comment regarding the discussion of individual differences in brain, behavior, and navigational strategies. Therefore, we have addressed this point accordingly in response #6.

- An important cue in navigation is motion parallax, as it can provide both depth and angular information and may be used for Euclidean triangulation. It seems that motion parallax information was present in the triangle completion experiments. Why was this not included in the model? At minimum I feel this source of information needs to be discussed and whether it would affect the results or not.

In particular the observation model/ observation variability (page 11, line 849-858) presumes that object or landmark information becomes more noisy with larger distances, and at more peripheral angles of the visual field.

During forward motion the opposite is true in some sense for motion parallax. A landmark that is infinitely far away provides the most reliable background for parallax information with an intermediate landmark/ object, and the angular change of the parallax cue for a given distance traveled is maximal towards the periphery of the visual field (max at 90° to the traveling direction). This seems to be at odds with the assumptions in the observation model.

Response #3:

Thank you for asking about one specific depth cue, as this points out that we need to be clearer about our modeling of sensory uncertainties. Before addressing the reviewer's specific point about motion parallax, we would like to clarify our central modeling assumptions, the level of detail in our modeling approach, the relationship between multiple cues to depth and the resulting integrated uncertainty, previously used observation models in navigation, and previous literature.

The observation model in the present work is neither concerned with a noisy percept of the retinal image of the observer (e.g., Vijayabaskaran & Cheng, 2022; Kang et al., 2023) nor does it consider all individual visual depth cues, depending on the reference between 13 and 18 different cues to depth, separately. Instead, we assumed that the navigator has access to an integrated percept based on different visual depth cues and an integrated multi-sensory signal of self-motion cues.

To highlight this modeling assumption we now added or changed the following sentences in the revised manuscript:

- *Importantly, we assume that the navigator has access to an integrated percept stemming from different visual depth cues, as well as an integrated multi-sensory signal of self-motion cues.* (pg. 4, lines 155-158)
- *The subjects' internal belief is continuously updated by an integrated sensory input from external cues and motor actions via an internal model of the underlying dynamics, which allows for inferring allocentric position from noisy egocentric sensory input and self-motion signals (Fig. 2b).* (pg. 4, lines 166-171)

In theory, the observation model could be extended to separately include multiple noisy depth cues varying in reliability. These cues could then be combined through perceptual cue

integration processes according to their respective reliabilities (Landy et al., 1995). Note that this would happen through the state estimation within the model, i.e., currently, the Extended Kalman Filter. Similarly, the self-motion cues available to the actor could be further categorized into various multi-sensory sources, such as optic flow, vestibular, and proprioceptive cues. These, too, could be integrated based on their respective reliability (Chrastil et al., 2019). Note, however, that this is not at all straightforward, as several depth cues are not independent, so that such an explicit model of the integration of multiple cues would need to take into account the degree of correlations between all pairs of individual cues and not much is known about all of these pairwise correlations in human perception (Oruç, Maloney, Landy, 2003).

A further complication is that motion parallax depends on the fixation location and head motion (e.g., Medendorp et al., 2003), i.e. given a particular position of a subject and a particular movement speed the motion parallax cues to the distances of objects depends on the fixated object, be it one of the targets or one of the landmarks. Thus, the relative reliabilities of motion parallax cues dynamically change with motion but also with each individual gaze shift and during locomotion with the ongoing vestibular ocular reflex, further complicating the analysis in terms of individual cues.

Consequently, a comprehensive understanding and explicit consideration of the impact of distinct cues related to self-motion and depth perception, along with their sequential integration during spatial navigation, remains an important area for future experimental investigations and modeling endeavors.

To reflect these points we now added the following sentences to the discussion section:

- *Additionally, given our primary focus on understanding how navigators combine noisy internal self-motion cues with noisy external visual cues to update their relative position in the environment and the dynamic interaction of these sources in forming uncertain internal representations, we did not treat individual contributions of different self-motion cues separately, such as vestibular or proprioceptive information [74], and visual depth cues, including motion parallax [75]. (pg. 10, lines 785-793)*
- *Prior research has established that both self-motion and visual depth cues are subject to Bayesian cue integration processes [78, 79]. This forms a basis for extending the current probabilistic model to support the integration of multiple depth and self-motion cues according to their relative reliabilities. (pg. 10, lines 799 - 803)*
- *Consequently, future behavioral and modeling work should address the explicit influence of these different cues and other sources of navigational variability specifically in the context of the triangle completion task. (pg. 10, lines 807 - 811)*

Given our assumption of an integrated depth percept, we also assumed that the reliability of distance perception would decrease with increased distance. Specifically, this assumption is based on the fact that most depth cues tend to decrease in reliability with increased distance (see Cutting & Vishton, 1995 and Brenner & Smeets, 2018, for two excellent reviews on this topic). This assumption is also supported by a large body of research on distance estimation using various response measures (see Daum & Hecht, 2009; Feldstein et al., 2020, for an

extensive review). The assumption of increased observation noise in the periphery of the visual field comes from the fact that visual acuity decreases with increasing eccentricity (e.g., Kerr 1971), and this decrease in resolution has been shown to affect the precision of size and distance perception of objects (e.g., Newsome, 1972).

Notably, both of these assumptions have also been used successfully in previous modeling work (Belousov et al., 2016; Mallot & Lancier, 2018), which we used to initially constrain the magnitude of model parameters relating to the scaling of perceptual noise across distances and peripheral angles in the visual field.

To reflect these central modeling assumptions of the observation model, we have now added the following passage, including additional citations to relevant work in the supplementary methods section (model parameters):

- *Importantly, observation noise is based on the fact that integrated depth cue reliability and distance estimation accuracy diminish with increased distance, as evidenced in the majority of depth cues [9–12], and increased noise in peripheral vision due to decreased visual acuity [13, 14]. The magnitude of these four parameters regarding the observation noise was constrained based on previous modeling work using similar observation models, in which the magnitude of the noise scales with the distance and direction away from the observer [7, 8].* (pg. 29, lines 1508 - 1512)

The effect of increased uncertainty in distance perception also seems evident from experimental data in the homing task by Zhao et al. 2015 – as the condition in which landmarks are moved further away (distal landmarks), while all other factors remain unchanged. This condition shows increased variability in walking endpoints, which in our model is attributed to perceptual factors relating to increased noise in observation with increasing landmark distance.

During forward motion the opposite is true in some sense for motion parallax. A landmark that is infinitely far away provides the most reliable background for parallax information with an intermediate landmark/ object, and the angular change of the parallax cue for a given distance traveled is maximal towards the periphery of the visual field (max at 90° to the traveling direction). This seems to be at odds with the assumptions in the observation model.

Yes, for an isolated motion parallax cue with constant fixation on the intermediate object, this describes the situation well. However, given our assumption of integrated depth and self-motion cues stated above and the scaling of distance errors based on the majority of depth cues, as well as the potentially shifting fixation locations during movement, we did not specifically consider motion parallax as a separate cue in our model.

We think that the reviewer raises a valid point and that it would be interesting to see whether different configurations of objects and landmarks leading to differences in motion parallax impact the shape and magnitude of empirically observed endpoint variability. However, this would require new experiments involving the manipulation of all individual depth cues' relative reliabilities, which is beyond the scope of the current work.

In the study of visual place-recognition models, where homing variability is exclusively attributed to perceptual processes, we are aware of two studies by Pickup et al. (2013) and Dreesbach et al. (2017). Both studies assessed the role of motion parallax in homing, demonstrating how variations in scene layouts and viewing position concerning three pole-like landmarks of consistent visual size altered the shape of subjects' error distribution. Nonetheless, these studies were based on different homing tasks, and they did not explicitly examine the effects of altering the landmarks' distance over extensive ranges and excluded numerous other important depth cues, which are present in the experiments we considered here.

To our knowledge, the effect of motion parallax (or different depth cues in general) has yet to be studied explicitly in the context of the triangle completion task with landmarks. Here, motion parallax could, in principle, be used during the outbound path of the homing task, where a goal location pole and landmarks in the background are visible simultaneously (see Zhao & Warren, 2015 – Figure 1b). In this case, we agree with the reviewer's statement that the effect of motion parallax would indeed be larger for more distal landmarks. However, motion parallax information should also be less reliable during homing when the post of the initial goal's location is no longer visible.

In order to quantify and incorporate the specific influence of motion parallax into our model, future work is required in which the reliability of depth and motion cues is systematically manipulated and assessed to understand their impact on observed patterns of endpoint variability.

Based on the reviewer's suggestion to specifically discuss the influence of motion parallax we added the following sentences in the discussion section:

- *Particularly, motion parallax is recognized as an important cue in visual place recognition models of homing behavior, where variations in scene layouts and the positioning of the viewer relative to three consistent-sized, pole-like landmarks have been shown to significantly influence the distribution of subjects' homing errors [76, 77]. (pg. 10, lines 793-798).*

- It is laudable that the authors explicitly examine the role of different sources of uncertainty. It would be helpful if more numerical reporting of the different types of uncertainties were reported, in given context and across contexts.

For example, at which relative uncertainty levels of sensory, motor, and representational noise do simulated or biological agents switch strategies?

How do the different types of uncertainty compare to each other in magnitude/ impact?

Response #4:

We thank the reviewer for their valuable input and for providing us with the opportunity to provide further clarification on these points.

Overall, to fully answer these questions will require extensive further simulations and analyses that go beyond the scope of this manuscript. However, the ablation studies (formerly named model comparison) and the additional random model comparisons that we now include in the manuscript, together with the additional discussion, provide the first answers to these questions, which we provide in the following.

How do the different types of uncertainty compare to each other in magnitude/ impact?

Throughout our manuscript, we have shown the discernible impact of the different uncertainties on the variability of our model's endpoints. However, it is crucial to recognize that our exploration and illustration of these uncertainties within and across (different) contexts is not exhaustive, given the practical and computational constraints inherent to such an endeavor.

Our analysis, which focuses on model ablations (previously termed model comparisons), where models with different uncertainty sources are compared, offers a quantitative insight into the effects of various uncertainties on endpoint variability (see Supplementary Figures 4,5,6,7). This method, which selectively excludes model components that correspond to different sources of uncertainty, reveals the complex and non-linear nature of these uncertainties and their interactions under diverse conditions and experimental manipulations.

Importantly, this analysis underscores that the impact of different uncertainty sources within and across different contexts is intricate and complex, instead of merely additive. Moreover, the uncertainty dynamically changes from time point to time point due to both the movement and the sensory information. Thus, to answer this question fully, we will need to characterize not only the fixed parameters of the model but also the dynamically changing internal beliefs during navigation and relate these to the observed navigation behavior. This is a major undertaking that warrants a separate study.

In response to Reviewer #2's request for clarity on specific model parameter choices, we have included an additional Supplementary Figure 3. This figure presents a detailed quantitative depiction of the scaling of different noise parameters and their associated uncertainties.

For example, at which relative uncertainty levels of sensory, motor, and representational noise do simulated or biological agents switch strategies?

We thank the reviewer for this question, which we acknowledge as being an important question. However, in the following we will delineate why an answer to this question is far from straightforward.

Looking at the literature on navigational strategies, in situations where multiple strategies are available, a significant challenge for navigators lies in arbitrating between these strategies effectively. Two basic propositions have been made, where navigators either switch between different navigation strategies/behaviors or integrate them (see Arleo & Rondi-Reig, 2007 and Parra-Barrero et al., 2023 for extensive discussions).

In Figure 2 of our manuscript, we show that in discrete scenarios that involve the presence or absence of particular cues and differences in the agent's level of knowledge give rise to different navigational strategies within the framework of optimal control under uncertainty. The observed different behaviors between strategies aligns with how these behaviors are typically described in the literature (Gallistel, 1990; Arleo & Rondi-Reig, 2007; Nyberg et al., 2022; Parra-Barrero et al., 2023) and explains why these discretely different strategies are often associated with differences in response variability (Foo & Warren, 2005).

However, again, the actions and, therefore, the observed strategy of the navigator arise from the internal beliefs which dynamically change from time step to time step and are a complex, nonlinear interaction of the external and internal uncertainties over space and time, which, to make things even more complicated, are actively shaped by the actions of the navigator. Thus, a characterization in terms of external uncertainties alone is limited. Instead, it is required to analyze a large number of simulations in which internal uncertainty data are collected.

Finally, while different navigational strategies, such as beaconing, vector-based navigation, and path integration, have been used throughout the literature to describe human behavior, it is by no means straightforward to unambiguously classify every navigational step as belonging to each of these discrete classes of strategies.

Another important issue that we have not discussed extensively pertains to the availability of multiple strategies. Consequently, we have added the following paragraph to the section "Navigational strategies emerge from optimal control's dynamic interactions of spatial uncertainties":

- *When faced with multiple cues and, therefore, multiple potential strategies, a considerable challenge lies in effectively arbitrating between these strategies. Previously, it has been suggested that navigators either switch between various navigation behaviors [19] or integrate them [40, 41]. Differences related to cue availability and the navigator's internal knowledge may primarily determine strategy switching behavior (as illustrated in Figure 2), whereas differences in uncertainties related to representational, motor, and perceptual processes mediate the integration of different strategies shaping the resultant navigational behavior and its variability. Intuitively, these uncertainties correspond to the fidelity of the navigating agent's allocentric knowledge of the surrounding environment (representation), their ability to update and maintain a sense of self-location in response to noisy actions (motor), and their ability to egocentrically perceive the distance and direction of objects in the environment required for both map-learning and self-localization (perceptual). In the presence of multiple cues and, therefore, potential strategies, differing levels of navigational variability may therefore emerge more subtly and continuously. Specifically, relative differences in uncertainties related to sensory, motor, and representational processes continuously shape the balance of how much cognitive map learning, self-localization and internal prediction from self-motion take place during navigation, which consequently affects navigational precision. (see pg. 4, lines 243 - 264)*

Because navigation is a partially observable task, where navigators lack direct access to their precise location in the environment and an accurate representation of their surroundings, they maintain a dynamic belief regarding these quantities. To update these beliefs and deal with the different sources of uncertainties, we employ probabilistic Bayesian filtering.

In Bayesian filtering, different sources of uncertainty shape how much specific parts of the belief state are affected in the prediction and update steps, respectively (see equations 10 and 12). Given the continuous nature of these belief-state innovations, statements about discrete switches in strategies are difficult in scenarios where multiple cues, such as landmarks and cues related to self-motion, are available. In these scenarios, the dynamic interaction of the different uncertainties continuously shapes the balance of how much cognitive map learning, self-localization, and internal prediction based on self-motion occurs during the update step.

For example, in navigation scenarios when the uncertainty in the map is relatively larger compared to the uncertainty in self-location, e.g., in an unfamiliar environment, agents initially rely relatively more on internal prediction to track their own position (path integration) and update their internal map accordingly by taking their current level of uncertainty into account (cognitive map learning). In contrast, if uncertainty in self-location is larger than the uncertainty in the internal map, e.g., when the agent is disoriented, agents determine their position using noisy observations in relation to the internal map (reorientation & landmark-based navigation).

Another phenomenon can also be seen in the experimental data, where navigating agents may rely less on visually perceived information from landmarks and more on prediction based on self-motion. Specifically, this occurs when landmark perception based on visual cues is less reliable than the internal prediction based on self-motion, as in the distal landmark experiment by Zhao et al. 2015. Intuitively, this makes sense because distance and directional information from landmarks that are 500 meters away may be less informative when trying to reach a target location precisely in the range of 2-4 meters. In this case, navigators would still perform landmark-based navigation, although less precise and relying more on internal predictions from self-motion instead of visually uncertain landmarks.

Thus, the model we present in this manuscript shows the strategies previously reported in the literature, particularly for specific scenarios such as beaconing when a target is visible or path integration when no landmarks are available. However, it can now be used in future research to systematically investigate how these strategies and trajectory as well as endpoint precision result from the complex interactions of internal beliefs. Indeed, we plan to do so with a combination of behavioral experiments and simulations specifically relating the behavior to the relative uncertainties of internal beliefs. However, this will require extensive experiments with many repeated trials from individual subjects and extensive modeling, which is beyond the scope of the current manuscript.

- Page 6 line 369. It is stated that the results allow to 'causally attribute the observed behavior' to the different, dynamically changing sources of uncertainty. What exactly do the authors base this on? Fitting the outcome of the 8 models to the experimental

data does, in my opinion, not warrant any causal claims. Correlation is not causation. Also see page 9, line 578.

A related point is that the manuscript does not discuss caveats adequately, for example the fact that the choices of models and parameters are necessarily limited. There may be, and probably are, model designs that would capture the behavior and data even better. This does not take away from the contribution of the current work, demonstrating how different sources of uncertainty can dynamically interact.

However, statements like 'This provides invaluable insights..' seem a bit exaggerated to me.

Response #5:

We thank the reviewer for raising this point and have now added clarification and adjusted our wording accordingly.

Our manuscript provides a generative Bayesian model that involves specific noise in perceptual, motor, and representational processes and demonstrates the resulting contributors to the observed endpoint variability. It is a Structural Causal Model in the sense of Judea Pearl; therefore, it is actually a causal model. However, we do not want to dwell on this point because it may elicit discussions that are not really the main focus of our current manuscript.

Importantly, our computational model serves as a generative model for the homing task. When analyzing the results of model simulations independently from experimental data, specific effects within our model can be considered causal when they respond to specific and isolated parameter changes or scenario variations, as demonstrated in Supplementary Figures 2, 4, 5, 6, and 7 in the context of model ablations (previously termed model comparison). These causal predictions based on model simulations become important for future work as they provide quantitative hypotheses that can be tested through carefully designed experiments and are then correlated with resultant behavior.

In light of this critical distinction and the reviewer's suggestion, we relaxed any causal claims concerning experimental data throughout the manuscript and toned down the overall language regarding causality. Specifically, we removed/rewritten the following sentences to address this point:

- *The errors and the variability of different navigation behaviors [14] are well predicted across multiple experiments and can be ~~causally~~ related to the dynamically interacting uncertainties and variabilities that accrue over space and time.* (see pg. 1, lines 82-86)
- *Here, we show in simulations of homing tasks across five experiments from three different studies and laboratories that the observed endpoint biases and variability is quantitatively predicted ~~and causally explained~~ by the interaction of three sources of sensory-motor noise, namely perceptual, representational, and motor noise when*

assuming that humans plan their actions based on subjective internal beliefs. (see pg. 6, lines 277-283)

- ~~*This allows for causally attributing observed behavior to the contributing and dynamically changing sources of uncertainties in goal-directed navigation.*~~

We also agree with the reviewer's statement that the model's limitations, as well as the choice of parameters, need to be discussed more. Therefore, we added an extensive paragraph discussing the limitations of our current approach and future work to our discussion section, which, among others, includes the following sentences:

- *Although the motion and observation models employed in our study are well-established in prior literature [5, 72], it is important to recognize that they may not fully capture every aspect of human navigational variability observed in the homing tasks.* (see pg. 10, lines 782-785)
- Additionally, other factors contributing to variability, such as individual differences, decision noise as reflected in occasional lapses from subjects, and potential trial-by-trial learning effects [79], remain to be explicitly incorporated into the model's formulation (see pg. 10, lines 803 - 807)

We have also changed our initial wording of 'invaluable insights' to 'novel opportunities':

- *This provides novel opportunities for future work in neuroscience aimed at relating neuronal activity to internal perceptions instead of external stimuli or states [85,86], and in the behavioral sciences, as it makes testable predictions for behavior in spatial navigation tasks [87].* (see pg. 11, lines 856-860)

- It seems that the authors suggest that their model(s) can fully explain the origin of different navigation strategies.

However, there is ample evidence that individual differences (across people) in navigation strategies are persistent to some degree, and are linked to different neural systems and brain regions (see for example Iaria et al. 2003 Cognitive Strategies Dependent on the Hippocampus and Caudate Nucleus in Human Navigation: Variability and Change with Practice, or Chersi & Burgess 2015 The cognitive architecture of spatial navigation: hippocampal and striatal contributions).

I feel the ample literature on individual differences in brain, behavior, and navigation strategy needs to be discussed, and what the implications are for the current work.

Response #6:

We thank the reviewer for raising this important point and entirely agree that there are differences between individuals regarding brain, behavior, and their use of different navigational strategies that need to be discussed. In fact, the present model is a

population-level model because the limited number of trials per subject in most data sets does not allow for consideration of inter-individual variations in model parameters.

In the sense of the levels of description by David Marr, we do not see a contradiction between computational models and implementational-level descriptions of behavior but see them as complementary. Thus, the present computational model allows relating individual differences in the navigation to specific parameters in the model and, hopefully, in the future, to their neuronal correlates. As a concrete example, it may be that subjects differ in the degree of internal model uncertainty and motor variability and that these differences can explain differences in their navigational strategies, which can reflect differences at the neuronal level.

However, another important aspect to highlight is that our analysis is exclusively dedicated to strategies known as “place-based strategies”, in contrast to those termed “response-based navigation strategies”.

Consequently, we have now rewritten the discussion paragraphs addressing the various navigation strategies:

- *Navigation strategies have broadly been categorized based on sensory cues, representation types, and computational mechanisms, mainly differentiating between response-based and place-based strategies [2, 41, 51]. These strategies are commonly associated with different neural systems, such as the hippocampal and striatal systems, and their adaptation has been suggested to be influenced by uncertainty-related factors [51, 52]. Here, we specifically focus on place-based navigational strategies within open-field tasks [19] involving continuous motor actions and sensory observations, in contrast to response-based maze tasks requiring discrete decision-making actions, which may become habitual through repetition [52]. (see pg. 10, lines 692 - 703)*
- *In this context, the homing task with landmarks elicits several navigation behaviors previously described as different and seemingly separate navigational strategies [3, 19–21]. We demonstrate that these strategies, whether they involve walking towards a visible target location (beaconing [18]), continuously updating self-location based on internally generated motor cues (path-integration [7]), navigating relative to an internal representation of space (vector-based navigation [11, 21], or orienting based on landmarks in the environment (landmark-based navigation [19]) are unified by the principle of optimal feedback control under perceptual, representational, and motor uncertainties. (see pg. 10, lines 704 - 714)*
- *Simulations of navigation behavior during triangle completion tasks reveal dynamic shifts between these strategies and their integration depending on the availability and relative uncertainty of different spatial cues. Importantly, this provides a computational-level explanation of why these seemingly different strategies are associated with different navigational biases and variability levels. For example, the presence of landmarks reduces motor errors and, thus, endpoint variability by orienting subjects in the environment [37] and continually providing visual feedback about their location, i.e., landmark-based navigation [19]. (see pg. 10, lines 714-720)*

We have also included a paragraph in our discussion section, where we specifically address interindividual differences in navigation strategies and variability in the context of our model and implications for current and future work:

- *Due to the limited number of trials per subject in most data sets, we did not consider inter-individual variations in model parameters. However, prior research has shown that there are important differences between individuals regarding their use of different strategies. These strategies are distinguished by factors such as navigational precision [19] or the types of errors individuals make [81]. Prior research has also found that differences in navigational precision are often reflected at the population level and change throughout development, continuing late into life [14, 82–84]. (see pgs. 10-11, lines 807-815)*
- *The present modeling framework provides researchers with the opportunity to generate predictions about the adoption and precision of different navigational strategies in the context of open-field navigation tasks. Specifically, it allows exploring how individuals use different strategies and this leads to differences in navigational variability. This variability may be related to internal uncertainties of perceptual, representational, and motor processes, alongside subjective behavioral costs that balance goal achievement, cognitive map learning, orientation, and effort minimization. (see pg. 11, lines 824-830)*
- *Therefore, individual differences in navigation behavior and variability may have a direct correspondence to parameters in our model, making inter-individual differences quantifiable in terms of differences related to subjective uncertainties and behavioral costs. However, addressing individual differences at this level is beyond the scope of the present work and should be investigated more closely in future work. (see pg. 11, lines 830-836)*

On a general level, differences between individuals' navigation behavior and response variabilities may stem from their ability to perceive external cues accurately (perceptual variability), update their internal estimates of self-position based on noisy actions and self-motion cues (motor variability), and learn, update, and maintain internal representations accurately (representational variability). Different subjective costs may also differentially shape an individual's behavior, as some individuals may be more prone to looking at landmarks to remain oriented, which may lead to less variable navigation outcomes (positional uncertainty cost). In contrast, other subjects emphasize reaching their navigational goal as fast or as efficiently as possible (goal cost vs. control cost), whereas others spend more time learning the environment's layout (map uncertainty cost).

Importantly, these kinds of quantitative predictions have now become possible for the first time using our computational framework. Currently, we are actively investigating these various aspects in follow-up experiments.

Minor:

- Please consider making the simulation scripts openly accessible.

Response #7:

Of course, code for running the model will be made available.

- **Figure 1d. Information missing on what the red and blue cross in the 'Proximity' panel represent (the mean?).**

Response #8:

Thank you for pointing this out!

The plot describes subjects' behavior in conflict conditions in which landmarks are shifted by 15° - accordingly the red and blue crosses refer to the location of the target if navigators were to rely either exclusively on self-motion (original target location) or landmarks (shifted target location).

We have now added the following sentence to the caption of Figure 1D to clarify this:

- *Red and blue crosses represent target locations based on exclusive reliance on either self-motion or landmarks* (see pg. 2 Figure caption 1D)

- **Figure 3 panel a. Information missing on what the red crosses represent. Also, there is a visibility issue with the red cross on top of red circles in the self-motion scatter plots.**

Response #9:

The red crosses consistently represent the normalized target location (0,0) across all plots and conditions, as illustrated in Figure 1a, where we've established the use of red to signify the target location. Importantly, the location of this normalized target is the same across all conditions and plots. However, to enhance the visibility of these crosses in the self-motion condition, we have now introduced a black outline in the corresponding plots.

To better align with the overall color scheme, we have also adjusted the crosses in Zhao et al. 2015 to a dark gray color in Supplementary Figures 5 and 7. This modification helps to clearly represent that these crosses indicate the normalized location of the third post, rather than the homing target location.

- page 22, line 1271: '..targets location' typo

- page 22, line 1273 '..targets locations' typo again?

Response #10:

Thank you for pointing these out! We have now fixed both typos in the revised manuscript accordingly.

References

- Vijayabaskaran, S., & Cheng, S. (2022). Navigation task and action space drive the emergence of egocentric and allocentric spatial representations. *PLOS Computational Biology*, 18(10), e1010320.
- Kang, Y. H., Wolpert, D. M., & Lengyel, M. (2023). Spatial uncertainty and environmental geometry in navigation. *bioRxiv*, 2023-01.
- Landy, M. S., Maloney, L. T., Johnston, E. B., & Young, M. (1995). Measurement and modeling of depth cue combination: in defense of weak fusion. *Vision research*, 35(3), 389-412
- Arechavaleta, G., Laumond, J. P., Hicheur, H., & Berthoz, A. (2008). An optimality principle governing human walking. *IEEE Transactions on Robotics*, 24(1), 5-14.
- Chrastil, E. R., Nicora, G. L., & Huang, A. (2019). Vision and proprioception make equal contributions to path integration in a novel homing task. *Cognition*, 192, 103998.
- Cutting, J. E., & Vishton, P. M. (1995). Perceiving layout and knowing distances: The integration, relative potency, and contextual use of different information about depth. In *Perception of space and motion* (pp. 69-117). Academic Press.
- Brenner, E., & Smeets, J. B. (2018). Depth perception. *Stevens' Handbook of Experimental Psychology and Cognitive Neuroscience*, 2, 1-30.
- Daum, S. O., & Hecht, H. (2009). Distance estimation in vista space. *Attention, Perception, & Psychophysics*, 71, 1127-1137.
- Feldstein, I. T., Kölsch, F. M., & Konrad, R. (2020). Egocentric distance perception: A comparative study investigating differences between real and virtual environments. *Perception*, 49(9), 940-967.
- Kerr, J. L. (1971). Visual resolution in the periphery. *Perception & Psychophysics*, 9, 375-378.
- Newsome, L. R. (1972). Visual angle and apparent size of objects in peripheral vision. *Perception & Psychophysics*, 12, 300-304.
- Medendorp, W. P., Tweed, D. B., & Crawford, J. D. (2003). Motion parallax is computed in the updating of human spatial memory. *Journal of Neuroscience*, 23(22), 8135-8142.
- Belousov, B., Neumann, G., Rothkopf, C. A., & Peters, J. R. (2016). Catching heuristics are optimal control policies. *Advances in neural information processing systems*, 29.
- Mallot, H. A., & Lancier, S. (2018). Place recognition from distant landmarks: human performance and maximum likelihood model. *Biological cybernetics*, 112(4), 291-303.
- Zhao, M., & Warren, W. H. (2015). How you get there from here: Interaction of visual landmarks and path integration in human navigation. *Psychological science*, 26(6), 915-924.
- Pickup, L. C., Fitzgibbon, A. W., & Glennerster, A. (2013). Modelling human visual navigation using multi-view scene reconstruction. *Biological cybernetics*, 107, 449-464.
- Gootjes-Dreesbach, L., Pickup, L. C., Fitzgibbon, A. W., & Glennerster, A. (2017). Comparison of view-based and reconstruction-based models of human navigational strategy. *Journal of vision*, 17(9),

11-11.

Nardini, M., Jones, P., Bedford, R., & Braddick, O. (2008). Development of cue integration in human navigation. *Current biology*, 18(9), 689-693.

Parra-Barrero, E., Vijayabaskaran, S., Seabrook, E., Wiskott, L., & Cheng, S. (2023). A Map of Spatial Navigation for Neuroscience. *Neuroscience & Biobehavioral Reviews*, 105200.

Gallistel, C. R. (1990). *The organization of learning*. The MIT Press.

Arleo, A., & Rondi-Reig, L. (2007). Multimodal sensory integration and concurrent navigation strategies for spatial cognition in real and artificial organisms. *Journal of integrative neuroscience*, 6(03), 327-366.

Nyberg, N., Duvelle, É., Barry, C., & Spiers, H. J. (2022). Spatial goal coding in the hippocampal formation. *Neuron*.

Foo, P., Warren, W. H., Duchon, A., & Tarr, M. J. (2005). Do Humans Integrate Routes Into a Cognitive Map? Map- Versus Landmark-Based Navigation of Novel Shortcuts. *Journal of Experimental Psychology: Learning, Memory, and Cognition*, 31(2), 195–215

Wolbers, T., & Hegarty, M. (2010). What determines our navigational abilities?. *Trends in cognitive sciences*, 14(3), 138-146.

Khamassi, M., & Humphries, M. D. (2012). Integrating cortico-limbic-basal ganglia architectures for learning model-based and model-free navigation strategies. *Frontiers in behavioral neuroscience*, 6, 79.

Chersi, F., & Burgess, N. (2015). The cognitive architecture of spatial navigation: hippocampal and striatal contributions. *Neuron*, 88(1), 64-77.

Loomis, J. M., Klatzky, R. L., Golledge, R. G., Cicinelli, J. G., Pellegrino, J. W., & Fry, P. A. (1993). Nonvisual navigation by blind and sighted: assessment of path integration ability. *Journal of Experimental Psychology: General*, 122(1), 73.

Iaria, G., Petrides, M., Dagher, A., Pike, B., & Bohbot, V. D. (2003). Cognitive strategies dependent on the hippocampus and caudate nucleus in human navigation: variability and change with practice. *Journal of Neuroscience*, 23(13), 5945-5952.

Bates, S. L., & Wolbers, T. (2014). How cognitive aging affects multisensory integration of navigational cues. *Neurobiology of aging*, 35(12), 2761-2769.

Petrini, K., Caradonna, A., Foster, C., Burgess, N., & Nardini, M. (2016). How vision and self-motion combine or compete during path reproduction changes with age. *Scientific reports*, 6(1), 29163.

Bostelmann, M., Lavenex, P., & Lavenex, P. B. (2020). Children five-to-nine years old can use path integration to build a cognitive map without vision. *Cognitive psychology*, 121, 101307.

Oruç, I., Maloney, L. T., & Landy, M. S. (2003). Weighted linear cue combination with possibly correlated error. *Vision research*, 43(23), 2451-2468.

Reviewer #2 (Remarks to the Author):

The manuscript presents several lines of results to support the bold hypothesis that patterns of endpoint bias and variability across multiple triangle completion experiments can be explained with a single dynamic Bayesian actor model of navigation.

The findings would be very interesting and noteworthy if the model fitting and comparison procedures were sound, but I have a few questions before I am reassured:

Response #1:

We greatly value the reviewer's feedback on our manuscript, and we are happy to provide additional background to our model and the procedures we used to constrain and fine-tune its parameters. These questions have led us to run a wealth of additional simulations and analyses, which we are now including in the revised version of the manuscript. We hope that these additional results address the reviewer's concerns satisfactorily.

First, our probabilistic Bayesian actor model follows a long tradition of models in motor control rooted in the principles of optimal feedback control under uncertainty (Todorov & Jordan, 2002; Wolpert & Ghahramani, 1995), a concept that has not yet been established in the field of spatial navigation.

One of the distinguishing features of our model lies in the fact that, in stark contrast to neural network models, which often involve idiosyncratic architectural choices and an overwhelming number of parameters, typically exceeding 100,000, the majority of our comparatively few parameters hold a clear and interpretable meaning and are therefore necessarily constrained in their overall magnitude to physically or biologically plausible ranges.

Not only does our model exhibit robustness and accuracy in capturing complex naturalistic goal-directed navigation behavior, but its parameters are also subject to the highest standard of evaluation through transfer testing, demonstrating that our model effectively generalizes across a wide range of experimental conditions, environmental settings and manipulations with a single set of parameters.

Following the reviewer's helpful suggestions, we have now extensively revised our manuscript by including new model simulations and analyses that illustrate the effects of various model parameters, and providing detailed justifications for our initial choice of these parameters. Details of these specific changes are found in the responses below.

Major:

1) Line 1251 states that the parameters were “manually fitted”. This seems to beg the question of how the 28 parameters listed in Table 2 could be optimized effectively.

Response #2:

We are sorry for having created the confusion and we thank the reviewer for raising this point, as it allows us to offer further and necessary clarification. We used the formulation “manually fitted” because we did not perform a global grid search or Bayesian optimization over all parameters but a combination of reasoning, literature research, and local optimization of individual parameters. In the following we will explain our reasoning and to address the reviewer’s subsequent related questions we will show in detail new simulations and analyses in the responses that follow.

Our model in the present manuscript provides a generative forward model, allowing us to simulate navigation behavior probabilistically across various goal-directed navigation tasks. Importantly, this means that no direct likelihood or gradient could be used for a straightforward algorithmic approach to model fitting.

Recent work from our lab has shown for the first time that likelihood-based model inversion for sequential visuomotor behavior is now possible for basic tasks involving non-linear motion and observation models (Straub et al., 2023). However, using such an approach is not feasible for the complex homing task and cost function we have used in the present manuscript.

Therefore, when faced with the problem of fitting the parameters systematically to empirically observed endpoint/response variability, we were initially presented with three possible approaches:

1. A basic grid search in a discretized 28-dimensional parameter space
2. Bayesian optimization or approximate Bayesian computation within a tightly constrained prior in a 28-dimensional parameter space
3. Manually constraining parameters in their magnitude based on previous literature and modeling work followed by fine-tuning to experimental data.

The computationally demanding nature of our model simulations becomes evident when considering that a single trial can take between 5-10 minutes on TU Darmstadt’s “Lichtenberg” high-performance compute cluster due to the complexity of belief-space planning with continuous states, non-linear actions and non-linear observations. These demands are further increased with changes in specific parameters, such as increases in the planning horizon, decreases in the control horizon, the number of landmarks in the environment, and belief-space planning which explicitly considers future uncertainties.

During the development of the model, we have spent significant time reducing overhead and speeding up model computations as much as possible to make the simulation of many trials feasible. However, to provide you with an estimate of the required compute time, any forward

simulation of a single set of parameters takes 120-180 minutes for all five experiments (a total of 5672 trials) when using 13 compute nodes on the TU Darmstadt high performance compute cluster, each node running 96 trials on two Intel® Xeon® Platinum Prozessor 9242 in parallel at any given point in time. It is important to note that the total CPU hours required for such simulations can vary significantly depending on the specifics of each experiment and parameter settings, typically ranging between 15 and 96 CPU hours. This is further complicated by stochasticity in the simulated model output, which would require additional forward simulation of the same parameter set to guarantee a robust fit.

Performing all experiments in the present manuscript before its revision (including ten runs for each of the eight different models, i.e., an additional 453.760 trials) for a single set of parameters may take up 50.000 CPU hours and three full days to compute. As most parameters are continuous, performing a grid search requires discretization into n steps. Thus, performing all possible combinations of parameter values would then scale the complexity to n^{28} runs. We may additionally mention that the simulations for the present revisions involved another 2 Mio. CPU hours.

While we wish to have a sufficiently fast model and compute resources to perform complete (or even Bayesian) optimization solely based on large-scale forward simulation, this is computationally not feasible, and developing such a methodology is beyond the scope of the present work. In either case, the objective would be to obtain physically, biologically, and cognitively plausible model parameters that lead to model behavior closely resembling human behavior across diverse experimental settings and manipulations.

Therefore, in principle, plausible ranges of model parameters, some owing to their physical nature, (1) are either directly reflected in human behavior (e.g., average walking or turning speed), (2) are amenable to careful measurement through tightly controlled psychophysical experiments, such as distance estimation or various locomotor tasks, (3) can be extracted from prior literature, modeling work, or experimental data.

This implies that only some parameter values necessarily require full optimization, whereas others can be constrained from the previously mentioned sources. Further, our model initially constrained by these sources and manually fitted to the experimental data by Nardini et. al 2008 is subjected to transfer testing, having to generalize across multiple experiments with different manipulations to the environment, which further constrains the range of possible models, specific parameter values, and their unique combinations.

The code of our model will of course be made available upon publication so readers can gain further intuitions regarding the complex interactions of different sources of noise and uncertainty during goal-directed navigation behavior.

Taken together, we currently do not have one single procedure to jointly optimize all 28 parameters listed in Table 2 effectively from scratch.

1a) It is mentioned that “noise parameters were constrained by findings from previous experiments”, but to me it was unclear whether this encompasses all or only some of motor, observation, and representation noise parameters. Clarifying exactly which ones were constrained by which experiment would be helpful.

Response #3:

We apologize for not having provided sufficient information in our original manuscript. In consequence, we now have substantially expanded the supplementary section on model parameters and Table 2 to provide information about which experiments were used to constrain the overall magnitude of noise parameters.

Importantly, given the limits on computation outlined in the previous response, the majority of these parameters were manually constrained offline using findings from previous experiments such that they are in physically, biologically, and cognitively plausible ranges and then manually tuned during model simulations to achieve the best possible fit for our population model for data by Nardini et al. 2008 only.

For each of the three different parameter types we have added the following sentences to the supplementary section (model parameters) explaining their role in the model as well as the specific assumptions, experimental and modeling work which was used by us to constrain their value ranges:

Observation noise parameters:

- *The observation noise parameters are responsible for errors the actor makes when observing landmarks at different distances and directions relative to the current viewing direction. They further shape the magnitude of observation uncertainty in the correction step, affecting how much positional information the actor can gain from a given landmark as well as shaping the uncertainty of landmarks in the internal map. Importantly noise in observation is based on the observation that depth cue reliability and distance estimation accuracy diminish with increased distance, as evidenced in the majority of depth cues [9–12], and the parameters also reflect increased noise in peripheral vision due to decreased visual acuity [13, 14]. The magnitude of these four parameters regarding the observation noise was constrained based on previous modeling work using similar observation models of landmarks/objects, in which the magnitude of the noise scales with the distance and direction away from the observer [7, 8]. (pg. 29, lines 1505 - 1512)*

Motor noise parameters

- *The motor noise parameters are responsible for signal-dependent errors in response to linear and angular velocity changes and directly shape the actor's internal positional and heading uncertainties via the prediction step. These capture any variability that is associated with the execution of motor actions, including changes of position through linear forward movement and changes of heading direction through turning of the body. (pg. 29, lines 1513 - 1516)*
- *The paper by Kallie et al. (2007) [4] provides basic estimates of step length variability (linear-linear velocity noise) and veering variability (linear-angular velocity noise) from a locomotor task (see Figure 8 and Table 5/6 of the paper), which we converted from step-based to velocity-based units. Both parameters were further adjusted to fit the*

velocity profiles and distributions of velocity present in the trajectory data from Chen et al. 2017 [3] and Zhao et al. 2015 [2]. Variability in the linear component from turning (angular-linear velocity noise), i.e., forward movement when turning on the spot, was set to be minimal, because the homing task did not necessarily require large turns, their impact can be considered negligible. Variability in turning (angular-angular velocity noise) was constrained in its overall magnitude using the paper by Jürgens et al. 1999 [6] (active turning condition AT). (pg. 29, lines 1516 - 1522)

Representation noise parameters:

- *The representation noise parameters influence the agent's uncertainty regarding internal estimates of landmark and object positions, which decay over time. Although the real-world dynamics for the navigation task do not include movement of objects or landmarks, we introduced this parameter to model the fact that agents cannot perfectly remember the position of landmarks and objects that they have previously encountered [18]. Therefore, the representation noise is best thought of as a process noise parameter increasing the uncertainty of the internal map representation over time. Given constraints on the other two parameters, we freely selected this parameter, assuming the variability would be equal in the x and y directions. Importantly, this representation noise remained constant throughout the experiment, because all trials were simulated identically and independent from each other and therefore we did not account for any trial-by-trial variability, e.g. due to learning. (pg. 29, lines 1523 - 1530)*

We also added a sentence highlighting the difference between internal uncertainties and those related to the generative process:

- *Importantly, we assumed these parameters to be equal for both the generative model of the task and subjects' internal uncertainties. (pg. 29, lines 1498 - 1499)*

Additionally, based on suggestions by Reviewer #1 we added a limitation paragraph in the discussion section discussing the limitation with regards to specific noise parameter choices:

- *Although the motion and observation models employed in our study are well-established in prior literature [5, 44], it is important to recognize that they may not fully capture every aspect of human navigational variability observed in the homing tasks. (pg. 10, lines 777 - 780)*
- *Additionally, other factors contributing to variability, such as individual differences, decision noise as reflected in occasional lapses from subjects, and potential trial-by-trial learning effects [80], remain to be explicitly incorporated into the model's formulation (pg. 10, lines 798 - 802)*
- *Due to the limited per subject data in most datasets, we did not consider potential inter-individual variations in navigation behavior. (pg. 10-11, lines 807 - 809)*

- However, addressing individual differences at this level is beyond the scope of the present work and should be investigated more closely in future work. (pg. 11, lines 829 - 831)

The overall effect of these parameters in terms of noise for different settings and at parameter scales of up to one order of magnitude higher than our initial selection is now illustrated in Supplementary Figure 3:

Supplementary Figure 3. Effect of parameter scales (1x, 2x, 5x, 10x) on motor, perceptual and representational variability: **a** Lateral deviation from walking straight with increasing walking distance for different scales of α_2 . **b** Variability in walking speed for a 1s walking bout as a function of walking speed for different scales of α_1 . **c** Average turning error for a 1-second turn at varying turning speeds at different parameter scales of α_3 . **d** Lateral deviation in position when turning on the spot for a 1-second turn for different scales of α_4 . **e** Average distance perception error for a centrally viewed object with increasing object distance for different parameter scales of σ_{rmin} . **f** Average distance perception error for a peripherally viewed object (3m distance) with increasing visual field angle σ_{rmax} at different parameter scales. **g** Average bearing perception error for increasing object distance for different parameter scales of $\sigma_{\psi min}$. **h** Average bearing perception error for peripherally viewed objects (5m distance) with increasing visual field angle for different parameter scales of $\sigma_{\psi max}$. **i** Representation error as a function of time for different parameter scales of δ_{xy} . **j** Parameter values of best-performing random parameter sets across different models for the experiment of Nardini et al. 2008.

The relative contribution and overall magnitude for our selected set of parameters to the observed endpoint variability is illustrated in Supplementary Figures 2, 4, 5, 6 and 7 in the context of model ablations (previously model comparison). We have also added extensive simulations studies and their analysis based on random model parameters which we will discuss in detail in the following response.

b) Even if all three kinds of noise parameters were constrained by previous experiments, it would still leave 19 parameters to optimize.

It would be more reassuring if it was shown that either the exact value of the parameters didn't matter (e.g., by performing grid search and showing the landscape of the loss function), or that the values were reasonable a priori.

For example, I wonder whether different values for the cost-function weights and/or the noise parameters not constrained by previous experiments (if any) would change the results qualitatively and make the other models (e.g., those without representation or motor noise) perform better.

Response #4:

We recognize and appreciate the reviewers' concerns regarding the number of parameters and the optimization process. As discussed in response #2, performing an exhaustive grid search or optimizing for all parameters simultaneously is practically infeasible due to computational constraints and time limitations.

Nevertheless, to address the concern raised by the reviewer, we have revised our manuscript by adding additional explanations regarding our assumptions behind specific parameter choices as well as additional simulations to describe the model's sensitivity to particular parameter choices.

Algorithm parameters:

Conducting 40 additional simulations for the experiment of Nardini et al. 2008 using varied parameters—specifically, exploring control horizons of 1, 5, 10, 15 and planning horizons of 5, 15, 20, 25, and 30 — we did not find significant changes in endpoint distributions across the different planning and control horizons (see Supplementary Figure 8 a and c).

An increased planning horizon will necessarily increase the computational demands of belief-space planning. Similarly, decreasing the control horizon means that the planning step is carried out more frequently in the model-predictive control loop, which also significantly increases the compute time of single trials. Consequently, the planning and control horizon was set primarily due to practical constraints on computation to make the simulations of many trials, experiments and the different models in the context of model ablations and model comparisons feasible (see Supplementary Figure 8 b and d).

To reflect these points we added the following sentences to our manuscript:

- *Conversely, the model showed less sensitivity to variations in planning and control horizons, as seen in Supplementary Figure 8. While adapting planning horizons is*

crucial in unpredictable environments and scenarios like uneven terrains [42] or obstacle avoidance [43], the homing task did not involve such unpredictability. (pg. 7, lines 414 - 419)

We also added an additional supplementary figure for these control analyses:

Supplementary Figure 8. Model comparison (planning and control horizons; cost-function) a Comparison of the influence of different planning horizons (0.5, 1.0, 1.5, 2.0, 2.5, 3.0 seconds) on endpoint distributions for the homing task of Nardini et al. 2008. Control horizon was fixed at 5 and dt was set to 0.1 seconds. 10 simulations per planning horizon. b Per-trial compute times for different planning horizons. c Comparison of the influence of different control horizons (0.1, 0.5, 1, 1.5 seconds) on endpoint distributions for the homing task of Nardini et al. 2008. Planning horizon was set fixed at 15 and dt was set to 0.1 seconds. 10 simulations per control horizon. d Per-trial compute times for different control horizons. e Random model comparison of 100 different cost-function weights vs. selected cost-function weights for Nardini et al. 2008.

Cost-functions weights:

The cost function weights were initially selected to produce behavior resembling those of human subjects found in trajectory data as close as possible (e.g., see Video S3 Human vs. Model Data and Supplementary Figure 10).

To assess the influence of different cost-function weights on trajectory walking endpoints, we performed an additional analysis by simulating 100 models for the experiment by Nardini et al. 2008 with randomly sampled cost-function weights (uniform sample between (0,10)). We compared them to the best-performing model under the selected parameters as reported in the manuscript.

Supplementary Figure 8e shows that the precise choice of cost-function weights in this range only moderately influence the model's ability to resemble endpoint variability across

the four experimental conditions. This aligns with existing research on similar human locomotor control models, as prior studies, like Arechavaleta et al. (2008) and Carlisle & Kuo (2023), have shown that differences in cost-functions and their weights mainly affect the shape and velocity profiles of walking trajectories when continual visual feedback is provided, rather than their endpoints. This aligns with our observations when we tuned cost parameters for walking trajectories. However, there are environments and navigation situations where the precise choice of these parameters is important, a topic we are currently examining through carefully designed experiments.

To reflect this point we added the following sentences to our manuscript:

- *Similarly, for the homing task by Nardini et al. 2008 the model was less sensitive to precise choice of cost-function weights (see Supplementary Figure 8e), complementing previous finding which show that difference in cost-function weights primarily influence the shape [44] and velocity profiles [45] of trajectories, but not necessarily their endpoints.* (pg. 7, lines 419-425):

Motion limits:

These parameters reflect basic properties of human walking, which in many cases are either well-established in the literature, human walking speed, turning speed, and their velocity profiles (Bohannon & Andrews, 2011; Carlisle & Kuo, 2023), or could also be obtained from analysis of trajectory data of the homing task. Therefore, our choice of these parameters constitutes a reasonable a-priori choice.

Initial belief state & initial uncertainty parameters:

We initialized the initial belief based on the assumption that agents performing the homing task do not know about the position of landmarks and goal location a-priori (i.e., high initial map uncertainty), which allows for identical independent simulation of trials.

Therefore, the agent built the internal map relative to their initial starting location (i.e., low initial position and heading uncertainty). Under this assumption, landmark and goal locations in the agent's belief were initialized relative to the position at which the agent encountered them using the current noisy observation. During navigation, these uncertainties are then influenced by relative differences in uncertainties related to motor, sensory, and representational processes, which the actor actively shapes through behavior.

The magnitude of variability and uncertainty in the agents' starting positions was derived from the starting positions for all trials in the trajectory data by Zhao et al. 2015. The effect of different initial uncertainties in self-position and the internal map, especially in the context of trial-by-trial learning, is also something we are currently starting to investigate more closely in our lab with dedicated experiments.

... For example, I wonder whether different values for the cost-function weights and/or the noise parameters not constrained by previous experiments (if any) would change the results qualitatively and make the other models (e.g., those without representation or motor noise) perform better...

In response to the reviewer's second point on whether other model variants, including different sources of noise, would perform better than a full model containing all sources, we have now performed additional analyses based on simulations of random noise parameter sets for all eight models.

This random model comparison based on randomly sampled parameter sets aims to explore the models expressiveness and the importance of particular parameter values:

- *To better understand the importance of particular noise sources in sensory, motor, and representational systems, we conducted additional simulations based on randomly sampled noise parameters, where each parameter was uniformly drawn between 0 and up to one magnitude higher than our selected parameter values (see Supplementary Figure 3 a-i). Model simulations using 1000 randomly sampled noise parameter sets reveal that the model is highly sensitive to the choice of particular noise parameters (see Supplementary Figure 9a) and that model simulation generating endpoint distributions more closely resembling empirically observed data are significantly influenced by particular noise parameters (see Supplementary Figure 9b). (pg. 6-7, lines 408 - 414)*

Supplementary Figure 9. Model comparison (random parameter sets) **a** Comparison of 1000 randomly sampled parameter sets versus selected model parameters for the homing task of Nardini et al. 2008. Random parameter sets were uniformly sampled between 0 and up to one magnitude higher of the selected parameter set. **b** GLM Analysis describing the impact of different noise parameters on the models ability to explain endpoint distribution for the homing task of Nardini et al. 2008. Asterisks denote significant correlations ($p < 0.05$). **c** Random parameter set model comparison with models containing different sources of variability and different numbers of noise parameters.

It is important to recognize that these different models exhibit variations in the number of free noise parameters, resulting in different levels of parameter space coverage when we sample random noise parameter sets. For instance, the model involving representation noise only has just one free noise parameter, while models involving either motor noise or perceptual noise have four parameters each. Because the full model with its nine different noise parameters offers the highest degree of expressiveness, it's unsurprising that we observe a wider range of errors across the randomly selected parameter sets.

In response to the reviewer's suggestion to test whether models excluding specific sources of variability would perform similar or comparatively better than the full model using our selected parameter set we have now added the following analysis:

- *Considering the potential significance of specific sources of uncertainty over others, we asked whether models with fewer sources of variability would provide similar or even superior predictions compared to a model that accounts for all sources of variability. However, due to the considerable computational demands of model simulations (requiring 5-10 minutes per trial), we could not perform an exhaustive model comparison where each model is separately optimized to provide the closest matching endpoint distributions. Therefore, we performed a random model comparison, where each of the eight models was simulated under 50 randomly sampled noise parameter sets. A closer analysis of energy distances for each of the eight models across the different experimental conditions suggests that some models are inherently limited in their expressiveness (Supplementary Figure 9c). For example, the zero, perceptual, and motor models show little to no variation in energy distances across the different conditions, irrespective of specific parameter values. Similarly, models that incorporate uncertainty in perception, representation, or both but lack motor variability are limited in their expressiveness in the conflict condition.*
(pg. 7, lines 426 - 445)

Based on the random simulations shown in Supplementary Figure 9c we selected the best-performing random parameter set for each model and subjected it to transfer testing. Transfer testing involves simulating data for all five experiments ten times each, allowing for a robust comparison against our selected parameter set.

First, to provide a clear understanding of the differences and similarities in parameters found using this procedure, we illustrated the relative scale of parameters compared to our chosen set in Supplementary Figure 2j.

Supplementary Figure 2. Effect of parameter scales on motor, perceptual and representational variability: ... j Parameter values of best-performing random parameter sets across different models for the experiment of Nardini et al. 2008

Second, we included an additional supplementary figure which illustrates differences between trajectories simulated under the different best-performing random parameter set for each model (see Supplementary Figure 10):

Supplementary Figure 10. Trajectories for Zhao et al 2015 proximal experiment: Human vs. Model data. Trajectories were simulated for the best-performing random parameter set for each of the different models.

Third, we included a plot which shows the transfer testing of the different models across all five experiments (see Supplementary Figure 11):

Supplementary Figure 10. Random model comparison across all experiments. For each of the eight different model the best-performing parameter sets in the task of Nardini et al. 2008 (Supplementary Figure ... c) were

selected and subjected to transfer testing across experiments from Zhao et al. 2015 and Chen et al. 2017 (10 simulations for each model).

The performance of specific models varied significantly across different experiments, with some models performing well in one experiment, but worse in others.

For instance, in the experiment by Nardini et al. 2008, where all landmarks and pickup objects are visible from the beginning, a model only considering motor variability fails to capture endpoint distributions compared to all other models except the zero and representation variability only models. However, the motor variability only model's predictions improve for both experiments by Chen et al. in 2017 and the proximal experiment by Zhao et al. in 2015, where goal post locations appear sequentially during the unfolding of the trial.

Furthermore, without perceptual variability, the motor variability-only model cannot account for the significant differences in endpoint distributions observed in Zhao et al.'s proximal and distal landmark conditions, making it perform worse than most other models. Similarly, the motor and representation variability model, which performs best in Zhao et al.'s 2015 proximal landmark experiment, demonstrates relatively poorer performance in the distal landmark experiment when compared to other models incorporating both perceptual and motor variability.

Taken together, this analysis reveals that while an expected improvement in certain homing conditions or experiments for the different models is to be expected, this does not guarantee the type of transfer across all conditions and experiments we have demonstrated.

Consequently, we addressed these points in the revised manuscript as follows:

- *Next, we chose the best-performing random parameter set for each model. We subjected them to transfer testing, where we simulated each model for all five experiments to assess their ability to generalize across these varied experimental environments and manipulations. For empirically observed endpoint variability, we found that while some models performed similarly or better than the model with our selected parameter set in certain experiments, overall, their generalization was inferior across all experiments and environments (see Supplementary Figure 11). Importantly, the parameters in the alternative models found using this procedure would require assigning noise parameters that are up to one order of magnitude higher compared to models excluding particular sources of uncertainty to compensate for the absence of variability in other sources. This leads to magnitudes that are physically, biologically, and cognitively rather implausible (see Supplementary Figure 3j). Additionally, for most models, this leads to walking trajectories that do not align well with empirically observed trajectories (see Supplementary Figure 10). (pg. 7, lines 446 - 462).*

2) Related to 1), I wonder whether parameters were optimized again when considering the cases when one or more of the motor, observation, or representation noise were

ignored (Fig 3h). If we optimize the parameters for the case when none of them are ignored, we can certainly expect the model to perform worse when some of them are ignored, due to overfitting.

Response #5:

We appreciate this valid concern raised by the reviewer and apologize for the confusion that may stem from our use of the term 'model comparison'.

In this context, we fully agree with the reviewer's statement concerning overfitting. Given the challenges of automatically optimizing the parameters, as discussed in response #2, we could not perform a classic model comparison in which separate models are independently fitted to achieve the best possible fit on a given dataset.

A model ablation study is a more accurate term for the method employed and the demonstration we sought through this analysis. Commonly used in machine learning contexts, this analysis aims to systematically remove specific model components to gain insights into the importance of different components and their interactions with other components.

This is similar to what we illustrated in Supplementary Figure 2 but in the context of the triangle completion task and across a broader range of different models, specifically those including two different sources of variability. With this analysis, we systematically assess the contribution of different sources of sensory-motor noise on the overall impact on endpoint variability in different experimental conditions and response to manipulations of the five experiments. To ensure the robustness of these ablations, we opted for ten simulations for all experiments and each of the eight models.

Based on the reviewer's suggestion related to response #4, we have now redone the "model comparison" (now "model ablation") from the previous version of the manuscript by eliminating the influence of noise in the generative process and setting the internal uncertainties close to zero accordingly.

Additionally, as illustrated in the previous response, we added a random model comparison, where we compared the best-performing models from a set of 50 randomly drawn parameter sets based on their ability to explain endpoint variability in the study by Nardini et al. 2008 and were then further subjected to transfer testing across all five experiments. Importantly, this analysis shows that when excluding specific sources of variability, variability in other sources is necessarily increased (see Straub & Rothkopf (2022) for a similar argument involving a basic object-tracking task involving perception and action).

3) It is unclear which results were used to fit the parameters and which results were predicted. In Line 269, it sounds like the parameters were fit to the results from

Nardini et al. 2008 and were used to predict results from Zhao et al. 2015 and Chen et al. 2015. However, in Line 1251, it is stated that the parameters were “fitted to trajectory and endpoint data for “all experiments simultaneously”. Which one was the case?

Response #6:

We thank the reviewer for raising this point and apologize again for any confusion arising from our unclear choice of wording.

After model parameters were initially constrained based on the steps outlined in previous responses, we prioritized fine-tuning the parameters to achieve the best possible fit to the data by Nardini et al. 2008 while staying within plausible parameter ranges. Importantly, trajectory data from the other two studies were initially used to extract plausible parameter values specifically for the homing task.

Here are again the details:

In our response #3 we detailed how the 9 noise parameters were set based on previous literature. Only the representational noise parameter, for which we could not find an appropriate empirical value in previous literature, was adjusted to yield a good match to the endpoint distributions in the Nardini et al. study.

In response #4 we detailed how the following parameters were set. The cost function weights were adjusted to resemble human navigation trajectories. However, the Nardini et al. data did not contain the full walking trajectories of subjects but only the trajectory endpoints. Therefore, we had to recur to behavioral data from Chen et al. 2017 and Zhao et al. 2015 that do contain full trajectories of individual participants to constrain the parameters resulting in plausible trajectories. Note that the additional simulations for this revision have revealed that the parameter range for the cost function weights producing a good fit to the trajectory endpoints is quite large. The planning algorithm parameters were selected so as to keep the computations feasible and were adjusted to match the Nardini et al. endpoint data. Note that the additional simulations for the revisions show that there is no major impact for different parameter settings in the range that we tried.

Taken together, the parameters were not fitted algorithmically but derived from a combination of reasoning, literature research, and local optimization, as discussed in the previous responses. Thus, we did not adjust the parameters to match the endpoint distributions for all datasets and we also did not adjust the parameters for each of the experiments separately. However, as there was no available trajectory data for the experiment by Nardini et al. 2008, some parameters, specifically those related to the cost function, were initially adjusted during model development based on trajectory data including the one available from Chen et al. 2017 and Zhao et al. 2015, but not to resemble the endpoint distributions, but only the trajectories.

Consequently, we have revised this in our manuscript accordingly:

- *Given the high computational demands of model simulations it was not possible to fit the model parameters to empirically observed endpoint data numerically. Instead, we used a combination of constraining parameters on the basis of previous research, e.g. the motor variability, or extracting parameter values from trajectory data, e.g. the average speed of walking, and finally by inspecting the endpoint distributions obtained from simulating our model. Initial values for all model parameters were chosen based on previous literature and sequentially adjusted manually so as to match the endpoint distributions in the Nardini et al. 2008 [1] study with the exception of the cost function parameters (see Supplementary Table 2 for parameter values). The reason is, that the Nardini et al. 2008 [1] data does not contain trajectory data but only endpoint distributions. The cost function parameters are known to strongly influence the shape of the trajectories [9, 12]. In order to set the cost function parameters initially, we therefore adjusted the cost function parameters in such a way that the model's walking trajectories resembled those from available data, i.e. Chen et al. 2017 [3] and Zhao et al. 2015 [13]. Note, that our additional analysis in section confirms that endpoint distributions did not change dramatically over a parameter range between 0 and 10 for each parameter when drawing random parameter values for the Nardini et al. 2008 study [1]. (pg. 29, lines 1498 - 1508).*

- *Similarly, the planning and control horizons did not change endpoint distributions drastically over a range of 0.5 to 3.0 seconds for the planning horizon and 0.5 to 1.5 seconds for the control horizon. Consequently, The planning and control horizon was set primarily due to practical constraints on computation to make the simulations of many trials, experiments and the different models in the context of model ablations and model comparisons feasible. Motion limit parameters were chosen based on established literature on human locomotion [14] in conjunction with extracted values from trajectory data by Zhao et al. 2015 [15] and Chen et. al 2017 [3]. Initial belief state and initial uncertainty were set based on the assumption that subjects did not know about the layout of the environment at the beginning of a trial. The initial positional uncertainty was extracted from trajectory data from the variance of starting locations. (pg. 29, lines 1509 - 1515).*

- *Importantly, to test whether the navigation model generalizes to navigation data from multiple other experimental settings and conditions across different labs, we used a single set of parameters for all simulations instead of adjusting parameters to each experiment or condition separately. This can be considered a form a transfer testing and the above results validate the model's ability to capture navigation data across triangle completion tasks. In the following we provide more detail about noise parameter settings individually. Importantly, we assumed that these parameters are equal for both the generative model of the task and the subjects' internal model. (pg. 29, lines 1516 - 1528).*

We hope that with the extended sections in the supplementary material that show model ablations, the behavior of models with random parameters, the comparison of parameter scales and the references to values reported in previous studies it is clearer how the parameters were selected.

Minor

4) Units need to be specified for parameters in Table 2.

Response #7:

We thank the reviewer for pointing out this oversight on our part.

We have now revised Table 2 to include units for model parameters whenever it was possible to do so. We also added an additional column indicating references to literature which we used to constrain the magnitude of model parameters related to our previous responses.

References

Todorov, E., & Jordan, M. I. (2002). Optimal feedback control as a theory of motor coordination. *Nature neuroscience*, 5(11), 1226-1235.

Wolpert, D. M., Ghahramani, Z., & Jordan, M. I. (1995). An internal model for sensorimotor integration. *Science*, 269(5232), 1880-1882.

Straub, D., Schultheis, M., Koepl, H., & Rothkopf, C. A. (2023, November). Probabilistic inverse optimal control for non-linear partially observable systems disentangles perceptual uncertainty and behavioral costs. In *Thirty-seventh Conference on Neural Information Processing Systems*.

Nardini, M., Jones, P., Bedford, R., & Braddick, O. (2008). Development of cue integration in human navigation. *Current biology*, 18(9), 689-693.

Kallie, C. S., Schrater, P. R., & Legge, G. E. (2007). Variability in stepping direction explains the veering behavior of blind walkers. *Journal of Experimental Psychology: Human Perception and Performance*, 33(1), 183.

Chen, X., McNamara, T. P., Kelly, J. W., & Wolbers, T. (2017). Cue combination in human spatial navigation. *Cognitive Psychology*, 95, 105-144.

Zhao, M., & Warren, W. H. (2015). How you get there from here: Interaction of visual landmarks and path integration in human navigation. *Psychological science*, 26(6), 915-924.

Jürgens, R., Boss, T., & Becker, W. (1999). Estimation of self-turning in the dark: comparison between active and passive rotation. *Experimental Brain Research*, 128, 491-504.

Mallot, H. A., & Lancier, S. (2018). Place recognition from distant landmarks: human performance and maximum likelihood model. *Biological cybernetics*, 112(4), 291-303.

Belousov, B., Neumann, G., Rothkopf, C. A., & Peters, J. R. (2016). Catching heuristics are optimal control policies. *Advances in neural information processing systems*, 29.

Fagan, W. F., Lewis, M. A., Auger-Méthé, M., Avgar, T., Benhamou, S., Breed, G., ... & Mueller, T. (2013). Spatial memory and animal movement. *Ecology letters*, 16(10), 1316-1329.

Darici, O., & Kuo, A. D. (2023). Humans plan for the near future to walk economically on uneven terrain. *Proceedings of the National Academy of Sciences*, 120(19), e2211405120.

Albrecht, S., Basili, P., Glasauer, S., Leibold, M., & Ulbrich, M. (2012). Modeling and analysis of human navigation with crossing interferer using inverse optimal control. *IFAC Proceedings Volumes*, 45(2), 475-480.

Straub, D., & Rothkopf, C. A. (2022). Putting perception into action with inverse optimal control for continuous psychophysics. *Elife*, 11, e76635.

Bohannon, R. W., & Andrews, A. W. (2011). Normal walking speed: a descriptive meta-analysis. *Physiotherapy*, 97(3), 182-189.

Carlisle, R. E., & Kuo, A. D. (2023). Optimization of energy and time predicts dynamic speeds for human walking. *Elife*, 12, e81939.

REVIEWERS' COMMENTS

Reviewer #1 (Remarks to the Author):

All my concerns have been addressed in the revised version of the manuscript. The additional analyses and results provide very helpful improvement to clarify the role of different sources of noise on the dynamic use of navigation strategies. The additional caveats and rewording align results and interpretations.

Reviewer #2 (Remarks to the Author):

I find the authors' response and revision reasonable. They convincingly describe why it is so difficult to perform parameter fitting in this domain. They now cite sources where parameters are set a priori, and show plots on the effects of the parameters where they don't matter much. They also provide results from random parameter search, which show, as they interpret, trajectories that look worse for most other models. Although representation or representation and motor models' trajectories look comparable to the full model, I agree with the authors that ignoring perceptual noise would be implausible and likely inconsistent with the literature. Combined, the revisions show that their approach was reasonable given practical limitations in computation.

Besides, I commend the authors for their plan to make their code publicly available. It will be a very valuable resource for the community.

In conclusion, I think the submission really pushes the boundary of the field, and I laud the authors for their effort, transparency, and comprehensiveness.

Reviewer #1 (Remarks to the Author):

All my concerns have been addressed in the revised version of the manuscript. The additional analyses and results provide very helpful improvement to clarify the role of different sources of noise on the dynamic use of navigation strategies. The additional caveats and rewording align results and interpretations.

Response #1:

Thank you for your feedback on the revisions and for your constructive comments in the initial review. We appreciate your acknowledgment of the improvements made in the manuscript.

Reviewer #2 (Remarks to the Author):

I find the authors' response and revision reasonable. They convincingly describe why it is so difficult to perform parameter fitting in this domain. They now cite sources where parameters are set a priori, and show plots on the effects of the parameters where they don't matter much. They also provide results from random parameter search, which show, as they interpret, trajectories that look worse for most other models. Although representation or representation and motor models' trajectories look comparable to the full model, I agree with the authors that ignoring perceptual noise would be implausible and likely inconsistent with the literature. Combined, the revisions show that their approach was reasonable given practical limitations in computation.

Besides, I commend the authors for their plan to make their code publicly available. It will be a very valuable resource for the community.

In conclusion, I think the submission really pushes the boundary of the field, and I laud the authors for their effort, transparency, and comprehensiveness.

Response #1:

Thank you for your thoughtful and detailed review. We are grateful for your recognition of the revisions and response efforts, particularly in addressing the complexities of parameter fitting.

We sincerely appreciate your positive evaluation of our work's contribution to the field.